# NetREX-CF integrates incomplete transcription factor data with gene expression to reconstruct gene regulatory networks

Yijie Wang [1,4✉], Hangnoh Lee [2,4], Justin M. Fear [2,4], Isabelle Berger[2], Brian Oliver [2✉] & Teresa M. Przytycka[3✉]

The inference of Gene Regulatory Networks (GRNs) is one of the key challenges in systems biology. Leading algorithms utilize, in addition to gene expression, prior knowledge such as Transcription Factor (TF) DNA binding motifs or results of TF binding experiments. However, such prior knowledge is typically incomplete, therefore, integrating it with gene expression to infer GRNs remains difficult. To address this challenge, we introduce NetREX-CF—Regulatory Network Reconstruction using EXpression and Collaborative Filtering—a GRN reconstruction approach that brings together Collaborative Filtering to address the incompleteness of the prior knowledge and a biologically justified model of gene expression (sparse Network Component Analysis based model). We validated the NetREX-CF using Yeast data and then used it to construct the GRN for *Drosophila Schneider* 2 (S2) cells. To corroborate the GRN, we performed a large-scale RNA-Seq analysis followed by a high-throughput RNAi treatment against all 465 expressed TFs in the cell line. Our knockdown result has not only extensively validated the GRN we built, but also provides a benchmark that our community can use for evaluating GRNs. Finally, we demonstrate that NetREX-CF can infer GRNs using single-cell RNA-Seq, and outperforms other methods, by using previously published human data.

[1] Computer Science Department, Indiana University, Bloomington, IN 47408, USA. [2] Laboratory of Cellular and Developmental Biology, National Institute of Diabetes and Digestive and Kidney Diseases, 50 South Drive, Bethesda, MD 20892, USA. [3] National Center of Biotechnology Information, National Library of Medicine, NIH, Bethesda, MD 20894, USA. [4] These authors contributed equally: Yijie Wang, Hangnoh Lee, Justin M. Fear. ✉email: yijwang@iu.edu; briano@niddk.nih.gov; przytyck@ncbi.nlm.nih.gov

Regulation of gene expression is central to cellular function. The regulatory relationships between transcription factors (TFs) and the genes they target are captured by the Gene Regulatory Network (GRN). Inference of these cell-type-specific GRNs is a current challenge in systems biology. Earlier work focused on predicting regulatory networks using gene expression data alone, but these methods tend to have poor predictive power[1,2]. Indeed, inference of network edges based solely on gene expression data is challenging[3,4]; network reconstruction uses an enormous search space, and the underlying biology is multi-layered with many factors including post-transcriptional and post-translation regulation contributing to TF's activity. We and others have found that network accuracy is drastically improved by including additional biological data such as chromatin structure (i.e., ATAC-Seq and ChIP-Seq), TF DNA binding motifs, and DNA sequence conservation scores[2,5–9].

Additional biological data have been used as a priori to inform network model selection in a variety of contexts[5,7,9–11]. MerlinP[7] uses network priors to influence the objective function for model selection. On the other hand, Inferelator[5], a method built on network component analysis (NCA), uses given gene expression data and a network prior to estimate TF activity. Furthermore, Inferelator predicts the GRN by uncovering the relationship between TF activity and their target genes' expression. We recently developed NetREX[2], which is also based on the NCA model, but NetREX simultaneously estimates TF activity while modifying the prior network by adding and removing edges.

Because of the NCA model's simplicity and biological relevance, this approach has become the foundation of the current state-of-the-art NCA-based methods for GRN reconstruction[2,5,6,12–16]. NCA uses the prior network's structure to inform the decomposition of gene expression into TF activities[12]. Specifically, TF activities are modeled as a hidden variable accounting for the complex and often unknown relationships between TF expression and TF regulatory activity. TF activity is more robust and has been proved to be superior to TF-gene expression in the task of GRN reconstruction[5]. However, NCA-based methods are more accurate when starting with a high-quality prior network. If a prior network is noisy, NCA-based methods cannot reliably predict TF activity, and in such circumstances the GRNs predicted by those methods are not trustworthy[2]. Therefore, building a reliable prior network becomes the key factor to employ the NCA-based methods.

A GRN prior is typically built by integrating various types of biological data, but the construction of a quality prior is challenging due to the incompleteness of available data. For example, we can build a prior network by using TF DNA binding data (e.g., ChIP-Seq). However, we often only have access to ChIP-Seq data for only a fraction of TFs. Therefore, all interactions with TFs that do not have ChIP-Seq data are considered missing values. Similarly, computational mapping of TF DNA binding motifs may miss true physical binding sites due to the problem of multiple testing, for example, leading to incompleteness in the TF DNA motif prior. Current methods for building GRNs by integrating multiple sources of prior knowledge do not directly account for the fact that there is missing data[9]. However, in the last decade, we have witnessed a rapid development of machine learning methods capable of dealing with large amounts of missing data. One particularly successful approach is Collaborative Filtering (CF), the method used by NETFLIX's movie recommendation system[17,18]. Given incomplete information about a user's preferences, CF infers informative features and then applies them to provide movie recommendations for other users in the absence of complete information. It has shown great potential in bioinformatics applications as well[19–21].

In this work, we present NetREX-CF—Regulatory Network Reconstruction using EXpression and Collaborative Filtering—a GRN reconstruction approach that uses the idea of CF, namely by combining such a recommendation system with expression-based model optimization. Similar to its precursor, NetREX, NetREX-CF selects a network model by simultaneously optimizing network topology and its NCA-based fit of gene expression data. However, rather than arriving at a final network by reprogramming the edges in the prior network, NetREX-CF uses a joint optimization function to directly integrate expression data with other types of prior knowledge using CF. We demonstrate that CF takes the fullest advantage of the prior data, and when combined with the biologically relevant NCA-based model, provided a remarkable improvement over existing approaches.

Mathematically, the simultaneous optimization of network topology, fit of the NCA model, and feature selection for the CF yielded a complex optimization problem of a type that has not been attempted before. The optimization is non-convex and non-smooth due to the binary nature of network edges (i.e., presence vs. absence). More importantly, the optimization contains $\ell_0$ norm that cannot be separated from other variables that need to be optimized. While the recently introduced Proximal Alternating Linearized Minimization (PALM) method[22] can solve a certain class of such non-convex optimization problems, where the $\ell_0$ norm is separable (in particular the one used in NetREX), simultaneous optimization of all three sets of parameters yields a problem that cannot be solved by PALM. To fill this gap, we introduce Generalized PALM (GPALM), a provably convergent method for solving a broad class of non-convex optimization problems with an inseparable $\ell_0$ norm.

In this study, we robustly tested NetREX-CF using our new data generated for this study as well as other studies that are based on yeast, fruit fly, and human cells. To start with, we validated the performance of NetREX-CF using public Yeast data. We compared NetREX-CF with previous leading approaches that use priors[2,6,7]. We use known TF-gene interactions as a benchmark set and apply Average Ranking Scores (ARS) to evaluate the performance of each competing method. The benchmarking results demonstrated that NetREX-CF significantly outperforms previous approaches. For additional validation of our method in multicellular and higher eukaryotic systems, we performed GRN construction using the fruit fly tissue-culture cell line as well as single-cell RNA-seq (scRNA-Seq) of human cells. First, we applied NetREX-CF to construct the GRN for the *Drosophila* Schneider 2 (S2) cell line. The S2 cell line is the key cell line used in experimental studies in this model organism[23–30], and PubMed lists over 11,000 papers with respect to the cell line. The construction of GRN for the S2 cell line not only demonstrates the power of our NetREX-CF but also the reconstructed network itself will be useful for guiding and interpreting future experiments. We collected S2 cell-specific gene expression, TF ChIP-Seq, and TF motif as the priors for the GRN construction. We also applied other leading approaches to infer the gene regulation in the S2 cell line. To validate the GRNs built by different methods, we experimentally generated and sequenced a total of 1920 RNA-Seq libraries following RNAi knockdown of 488 genes that encode all expressed TFs in *Drosophila* S2 tissue-culture cells. We then used the results of our RNAi gene knockdown experiment as the benchmark dataset to evaluate the GRNs that are computationally inferred. Lastly, we analyzed human scRNA-Seq data to illustrate that NetREX-CF can be applied to the scRNA-Seq outcomes. We generated cell-type-specific GRNs for human hepatocyte-like cells and embryonic stem cells[31,32] using functional prior and non-specific prior. We took the advantage of cell-type-specific information from public databases (e.g., ChIP-Seq) to produce benchmark datasets for our evaluation. From all the examples that we tested, NetREX-CF significantly outperformed previous approaches.

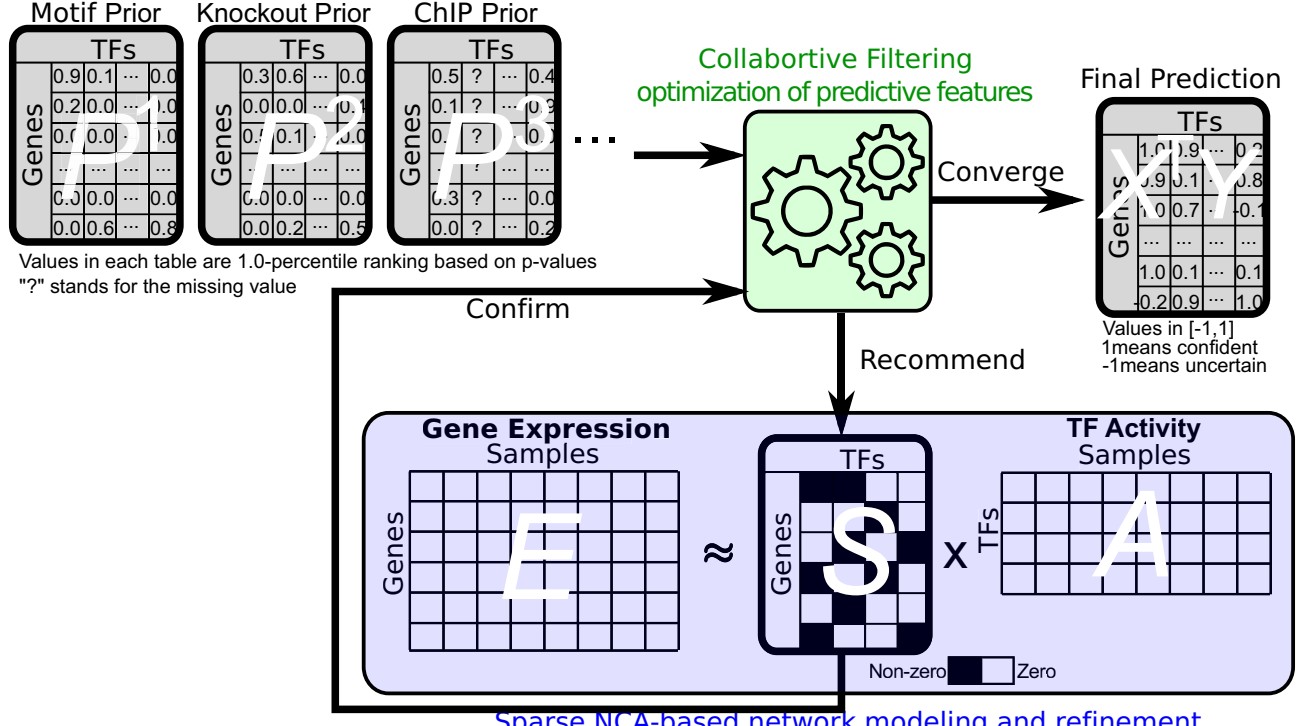

**Fig. 1 Method overview.** Collaborative filtering (CF) and NCA-based gene expression modeling alternatively inform each other during a joint optimization process: CF recommends edges to be confirmed by the NCA model and used to improve CF.

## Results

**NetREX-CF—method overview.** The NetREX-CF model is a data integration framework for reconstructing GRNs by organically utilizing both gene expression $E$ and a set of prior networks $P = \{P^1, \ldots P^{pd}\}$. The main idea behind the NetREX-CF model is an integration of two complementary optimization strategies: (i) a machine learning component designed based on CF that is able to infer hidden features from the current observed prior networks $P$ and utilize these features to recommend an improved GRN and (ii) a sparse NCA-based network remodeling component that can refine the topology of a GRN based on given gene expression $E$. These two computational components operate alternatively. The CF component recommends new edges to the current GRN and the sparse NCA-based network remodeling component screens the recommended edges and keeps the edges that are essential to explain the expression of a given gene. Once the sparse NCA-based network remodeling component confirms some of the recommended edges, the CF component further utilizes those retained recommended edges to make new edge recommendations for the sparse NCA-based network remodeling component to further examine (illustrated in Fig. 1).

Computationally, the system illustrated in Fig. 1 is achieved by simultaneous optimization of the following sets of variables: (i) the activities of TFs (matrix $A$), (ii) a weighted GRN (matrix $S$), and (iii) two feature matrices: the hidden features for target genes ($X$ where the $i$th row $x_i$ represents the hidden feature vector for gene $i$) and the hidden features for TFs ($Y$ where the $j$th row $y_j$ represents the hidden feature vector for TF $j$). The matrix $A$ is optimized by the sparse NCA-based network remodeling component while the matrices $X$ and $Y$ are optimized by the CF component. Notably, the matrix $S$ is the connection between the aforementioned two components and should be optimized by considering both components.

Formally, $E \in \mathbb{R}^{n \times l}$ is the matrix of expression data of $n$ genes in $l$ experiments and prior network $P^k \in \mathbb{R}^{n \times m}$, $\forall k$ is a weighted

adjacency matrix of the bipartite graph that records the prior knowledge of regulations between $m$ TFs and $n$ genes. Matrix $A \in \mathbb{R}^{m \times l}$ is the TF activity for $m$ TFs in $l$ samples and $S \in \mathbb{R}^{n \times m}$ is a weighted GRN. We further define penalty matrix $C$ and observation matrix $B$ based on the set of prior networks $P$. The matrix $C$ is used in CF component. For edges in the prior, the corresponding elements in $C$ will assign larger values to make sure those edges will be kept in the final prediction. For edges not in the prior, the corresponding elements in $C$ will assign smaller values to encourage new edges as recommendations. The matrix $B$ is used to indicate which edges have prior information. Each element in $C$ can be computed by $C_{ij} = 1 + a\sum_k P_{ij}^k$ ($a = 60$ suggested by ref. [18]). If more than one prior network suggests the regulation between the $i$th gene and the $j$th TF, then $C_{ij}$ tends to have a larger value. Large $C_{ij}$ would enforce giving the regulation between the $i$th gene and the $j$th TF a lower ranking. Each element in $B$ is binary and can be computed by $B_{ij} = 1$ if $\sum_k P_{ij}^k \neq 0$ and $B_{ij} = 0$ otherwise. $X \in \mathbb{R}^{n \times h}$ contains feature vector $x_i$ for gene $i$ and $Y \in \mathbb{R}^{m \times h}$ contains feature vector $y_j$ for TF $j$. Then, our optimization problem is formalized as following:

$$\min_{S,A,X,Y} \quad \mathcal{H}(S,A) + \lambda \mathcal{F}(S,X,Y)$$
$$s.t. \quad \|x_i\|^2 \leq 1, \forall i \qquad (1)$$
$$\|y_j\|^2 \leq 1, \forall j.$$

where $\mathcal{H}(S,A) := \|E - SA\|_F^2 + \lambda_A \|A\|_F^2 + \lambda_S \|S\|_F^2 + \sum_{ij} \eta_{ij} \|S_{ij}\|_0$ is the sparse NCA-based network remodeling component; $\lambda_A \|A\|_F^2 + \lambda_S \|S\|_F^2$ are standard regularization terms and $\sum_{ij} \eta_{ij} \|S_{ij}\|_0$ induces sparsity of a given prior GRN and therefore only essential edges that help to minimize $\mathcal{H}(S,A)$ are retained. $\|S_{ij}\|_0$ is the $\ell_0$ norm that is 1 if $S_{ij} \neq 0$ and 0 otherwise.

In (1) $\mathcal{F}(S,X,Y) := \sum_{i,j} \Omega_{ij} (\Theta_{ij} - x_i^T y_j)^2$ optimizes the hidden features $X$ and $Y$ of the CF component; $\Theta_{ij}$ is a binary matrix of

**Table 1 Overlap between prior networks and the gold standard network.**

| Network | No. of genes | No. of TFs | No. of Edges | No. of overlap with motif | No. of overlap with knockdown | No. of overlap with ChIP | No. of overlap with YEASTRACT |
|---|---|---|---|---|---|---|---|
| Motif | 5506 | 197 | 187,079 | 187,079 (100%) | 9236 (8.4%) | 8717 (3.5%) | 3497 (31.0%) |
| knockdown | 5543 | 262 | 96,809 | 9236 (4.6%) | 96, 809 (100%) | 7027 (2.9%) | 3050 (27.0%) |
| ChIP | 5557 | 318 | 229,936 | 8717 (4.3%) | 7027 (6.4%) | 229,936 (100%) | 2656 (23.5%) |
| YEASTRACT | 3731 | 148 | 10,525 | 3497 (1.7%) | 3050 (2.8%) | 2656 (1.1%) | 10,525 (100%) |

The last four columns show overlap between different networks and parentheses show the corresponding percentage. YEASTRACT is the gold standard network we used for performance comparison.

edges to be predicted by the hidden features in the given iteration and $\Omega_{ij}$ encodes penalties that guide the predictions. Both $\Theta_{ij} := \|S_{ij}\|_0 \oplus B_{ij}$ and $\Omega_{ij} := \bar{C}_{ij}\|S_{ij}\|_0 + C_{ij}(1 - \|S_{ij}\|_0)$ are defined based on $\|S_{ij}\|_0$ and the penalty matrix $C_{ij}$ is built from the prior information . Detailed explanation of $\Theta_{ij}$ and $\Omega_{ij}$ are provided in Method Details section. For the initialization step, both $\Theta_{ij}$ and $\Omega_{ij}$ are defined based on the prior networks only while in the subsequent steps they also take into account the results of the sparse NCA-based network remodeling component (illustrated in Fig. 1). To solve the problem (1), we first put all continuous terms together and define $H(S, A) := \|E - SA\|_F^2 + \lambda_A\|A\|_F^2 + \lambda_S\|S\|_F^2$ and then put all the non-continuous terms together and define $F(S, X, Y) := \sum_{i,j}\Omega_{ij}(\|S_{ij}\|_0 \oplus B_{ij} - x_i^T y_j)^2 + \sum_{ij}\eta_{ij}\|S_{ij}\|_0$. Then the optimization problem has a general format of an objective function as $\Phi(S, A, X, Y) = H(S, A) + F(S, X, Y)$, where $H(S, A)$ is continuous but non-convex and $F(S, X, Y)$ is a composite function of $\ell_0$ norm of elements of $S$ and other variables so it is neither continuous nor convex. More importantly, $\|S_{ij}\|_0$ is coupled with $x_i$ and $y_j$, so that $\|S_{ij}\|_0$ cannot be separated from $F(S, X, Y)$ as a distinct term. To the best of our knowledge, there has been no known method that can optimize such a complex and non-convex function involving the inseparable $\ell_0$ norm. To fill this gap, we propose here an algorithm, GPALM that generalizes the so-called PALM method[22] and solves a class of problems of the format above, under the assumption that $F(S, X, Y)$ is lower semi-continuous (see Supplementary Note 1). The GPALM method is fully described in Supplementary Note 2 where we also formally prove its convergence. The source code of NetREX-CF is available at: https://github.com/EJIUB/NetREX_CF.

**Validation and benchmarking NetREX-CF on yeast data.** To demonstrate the capability of our GRN reconstruction method, we tested using datasets from multiple species, which include yeast, fruit fly, and human. For yeast, we collect multiple datasets that measure different perspectives of gene regulation. These datasets include TF ChIP[7,33,34], TF DNA binding motif[7,35], genetic knockdown[7,36,37], and yeast gene expression[7,38–40]. TF ChIP, motif, and genetic knockdown datasets, serving as prior knowledge for TF-gene interactions in the yeast GRN. The details of these priors are summarized in Table 1 and the overlap among priors is illustrated in Table 1. We further utilize TF-gene interactions extracted from YEASTRACT database[41] as a gold standard to validate the performance of GRN reconstruction. These gold standard TF-gene interactions are supported by both DNA binding and expression evidence. The details of the gold standard TF-gene interactions and their overlap with the prior datasets are shown in Table 1. Results generated by NetREX-CF are benchmarked against the results obtained from the published sequential methods, all of which are GRN prediction methods that are able to use prior knowledge. In the sections that follow, we go into detail about the comparisons between two popular approaches that only consider gene expression, GENIE3[42] and GRNBoost2[43],

as well as prior-based approaches like NetREX-CF, MerlinP[7], NetREX[2], LassoStARS[6], the original CF[18], the summation of all prior knowledge (PriorSum), and a technique that assigns a random confidence score (uniformly distributed between 0 and 1; hereafter, a random method). For a detailed description of parameter selection for competing methods, we refer the reader to Supplementary Note 3.

To ensure an impartial comparison, we use Average Rank scores (ARS)[18]. For each method and for each gene $i$, we can obtain a list of TFs that are predicted to regulate gene $i$ and sort this TFs by the confidence of the prediction (most confident have higher rank). We use $r_{ij}^g$ to denote the percentile-ranking of TF $j$ within the ordered list of all TFs for gene $i$. Thus, $r_{ij}^g = 0\%$ means that TF $j$ is predicted with the highest confidence to regulate gene $i$, preceding all other TFs in the list. On the other hand, $r_{ij}^g = 100\%$ when TF $j$ is predicted to be the least possible TF for gene $i$ or there is no prediction between TF $j$ and gene $i$ yielded by the method. Based on the gold standard TF-gene interaction dataset $I$, we set $I_{ij} = 1$ if TF $j$ regulates gene $i$ in the gold standard dataset and $I_{ij} = 0$ otherwise. For any gene $i$, we use the average rank of the gold standard edges in the list of TF predicted to regulate gene $i$ as the measure quality of the prediction:

$$\overline{rank}_i^g = \frac{\sum_j r_{ij}^g I_{ij}}{\sum_j I_{ij}} \quad (2)$$

Lower values of $\overline{rank}_i^g$ are preferable, as they indicate gold standard TFs for gene $i$ have a lower rank than others. Furthermore, the overall ranking considering all genes can be computed by

$$\overline{rank}^g = \frac{\sum_i \overline{rank}_i^g}{\#\text{genes in } I} \quad (3)$$

The denominator is the number of genes that have gold standard TFs in dataset $I$.

Similarly, for each TF we can measure the quality of the sorted list of genes predicted to be regulated by it:

$$\overline{rank}_j^t = \frac{\sum_i r_{ij}^t I_{ij}}{\sum_i I_{ij}}, \quad \overline{rank}^t = \frac{\sum_j \overline{rank}_j^t}{\#\text{TFs in } I}, \quad (4)$$

where $r_{ij}^t$ denotes the percentile-ranking of gene $i$ within the ordered list of all genes for TF $j$ and $\overline{rank}_j^t$ is the average rankings for the gold standard genes for TF $j$. $\overline{rank}^t$ is the overall average rankings considering all TFs.

Figure 2a illustrates the comparison between the competing algorithms in terms of average rankings of gold standard TFs for each target gene. The sorted average ranking curve for NetREX-CF is below all other methods, indicating that the average rankings of gold standard TFs predicted by NetREX-CF for each gene are much lower than the rankings predicted by other methods. In the average rankings of gold standard genes among the genes predicted to be regulated by each TF, NetREX-CF still

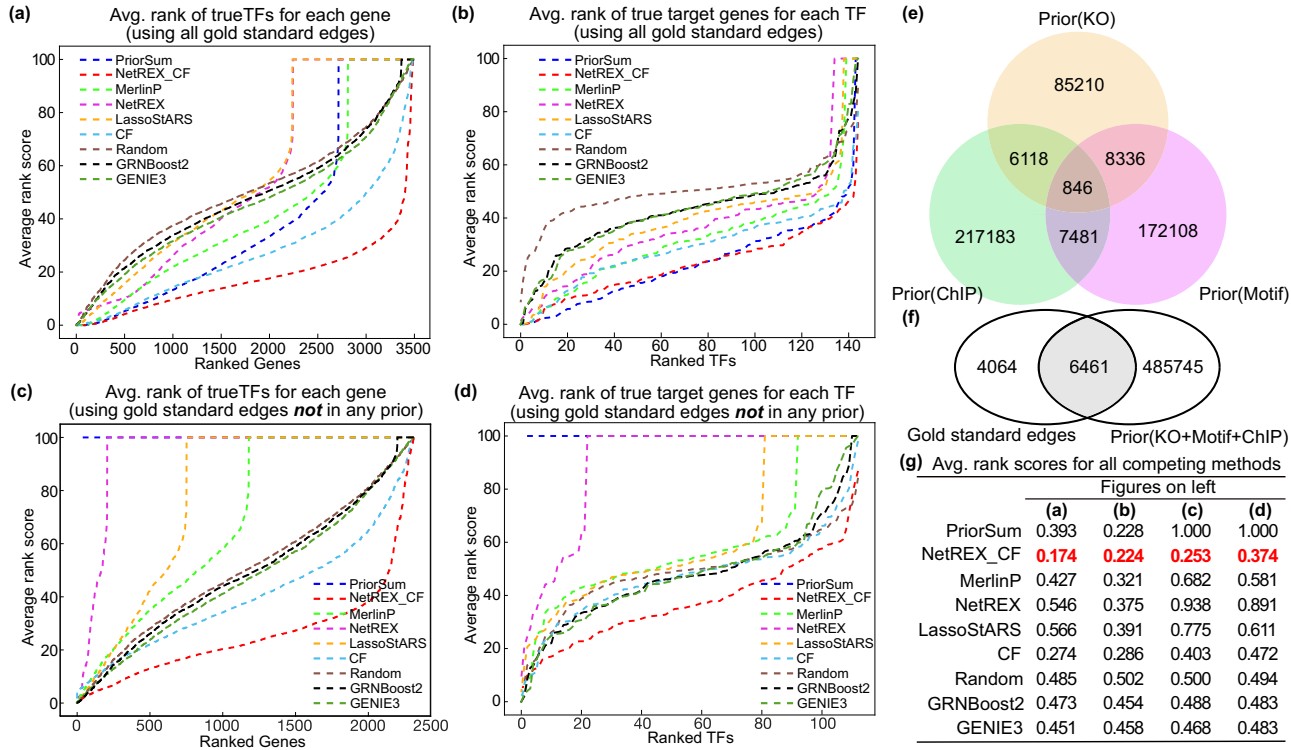

**Fig. 2 Performance comparison for all competing algorithms on the yeast dataset. a** The performance of the methods on the task of predicting regulating TFs: for each algorithm, we compute for each gene the rankings of gold standard edges $\overline{rank}_i^g$ adjacent to it, sort them in descending order, and plot the sorted average rankings. **b** The performance of the methods on the task of predicting regulated genes using a measure similar to in (**a**) but focusing on genes regulated by TFs. **c** The performance of the methods on the task of predicting regulating TFs that are not observed in prior data. The procedure is the same as in (**a**) but only the gold standard edges that are not included in the prior knowledge are used for the evaluation. **d** The performance of the methods on the task of predicting regulated genes not observed in prior data (similar to (**c**) but focusing on genes regulated by TFs). **e** Overlap between priors. **f** Venn diagram for gold standard dataset and the union of the three prior datasets. **g** Summary of the average rankings of each algorithm for tasks reported in panels (**a–d**). The lowest scores are highlighted in bold red.

achieves the best average ranking score. Remarkably, PriorSum (the weighted edge summation of three priors, ARS = 0.228) is very competitive with NetREX-CF (Fig. 2b, ARS = 0.224) indicating that NetREX-CF takes the best advantage of the prior data. It is important to note that NetREX-CF outperforms the original CF (ARS = 0.288), which demonstrates that the integration of the CF model and sparse NCA-based model is beneficial.

Next, in order to demonstrate the advantages of NetREX-CF in predicting ranks for missing data (edges that do not appear in the prior knowledge datasets), we identified all edges that are in the gold standard dataset but are not supported by any prior dataset. Indeed, a large portion of the gold standard dataset (4064 out of 10,525 gold standard TF-gene interactions) is not covered by any prior dataset (Fig. 2f). Therefore, we can use these gold standard interactions with missing prior data to compare the ability of the competing methods in recovering rankings under the assumption of missing data. NetREX-CF achieves much lower ARS for those missing data (Fig. 2c, d). In summary, NetREX-CF achieves the lowest, thus the best, overall ARS (Fig. 2g).

**Reconstruction of the GRN for *Drosophila* Schneider 2 (S2) cell line and validation of NetREX-CF on new experimental data.** While the datasets from yeast have been widely served as the only available benchmarking dataset for the evaluation of GRN inference methods, evaluating any GRN inference method using the only case would not produce convincing results. Therefore, for a more comprehensive assessment of network inference by

NetREX-CF, we have performed gene expression profiling of *Drosophila* S2 cells, in which we selectively silenced each known TF. We chose S2 cells for these reasons. First, the *Drosophila* community extensively uses S2 cells, and there exists a huge amount of public resources. Such a resource includes TF binding profiling (i.e., ChIP-chip/Seq), performed by the Model Organism ENCyclopedia Of DNA Elements consortium (modENCODE[44]). Second, the cells are suitable for high-throughput RNAi studies[45], whose outcome can be used to construct a benchmarking dataset when combined with genome-wide gene expression profiling.

As a prior, we collected publicly available datasets for S2 cells, which include 250 TF ChIP and 600 expression profiles (see Methods). TF DNA binding motifs were obtained from OnTheFly[46], FlyFactorSurvey[47], FLYREG v2[48], iDMMPMM[49], and DMMPMM[50]. Similar to the yeast GRN, we used TF ChIP and TF DNA binding motifs as prior knowledge for TF-gene interactions in the S2 cell GRN. In addition, we also included the gene co-expression network (from 600 expression profiles) to increase the overall number of TFs with prior information (Table 2).

In order to validate the performance of our NetREX-CF and also to provide the finest dataset for GRN inference evaluation, we experimentally generated and sequenced a total of 1920 RNA-Seq libraries following RNAi knockdown of 488 genes (465 after filtering, see the Methods) that encode all expressed TFs in *Drosophila* S2 tissue-culture cells. Since this is a new dataset that is likely to be valuable outside the current study, we start with a robust analysis of the properties of the data. We used long double-strand RNA molecules (dsRNA) to selectively silence

**Table 2 Overlap between prior networks and the benchmarking network for *Drosophila* S2 cell line.**

| Network | No. of genes | No. of TFs | No. of edges | No. of overlap with motif | No. of overlap with ChIP | No. of overlap with correlation | No. of overlap with knockdown |
|---|---|---|---|---|---|---|---|
| Motif | 6044 | 99 | 287,847 | 287,847 (100%) | 6806 (6.9%) | 59,084 (10.9%) | 3112 (6.2%) |
| ChIP | 6996 | 25 | 98,970 | 6806 (6.9%) | 98,970 (100%) | 17,990 (3.3%) | 3047 (6.1%) |
| Correlation | 7034 | 457 | 544,485 | 59,084 (10.9%) | 17,990 (3.3%) | 544,485 (100%) | 6994 (13.9%) |
| Knockdown (gold standard) | 7034 | 457 | 50,224 | 3112 (6.2%) | 3047 (6.1%) | 6994 (13.9%) | 50,224 (100%) |

The last four columns show overlap between different networks and parentheses show the corresponding percentage. Knockdown is the benchmark dataset we used for performance comparison.

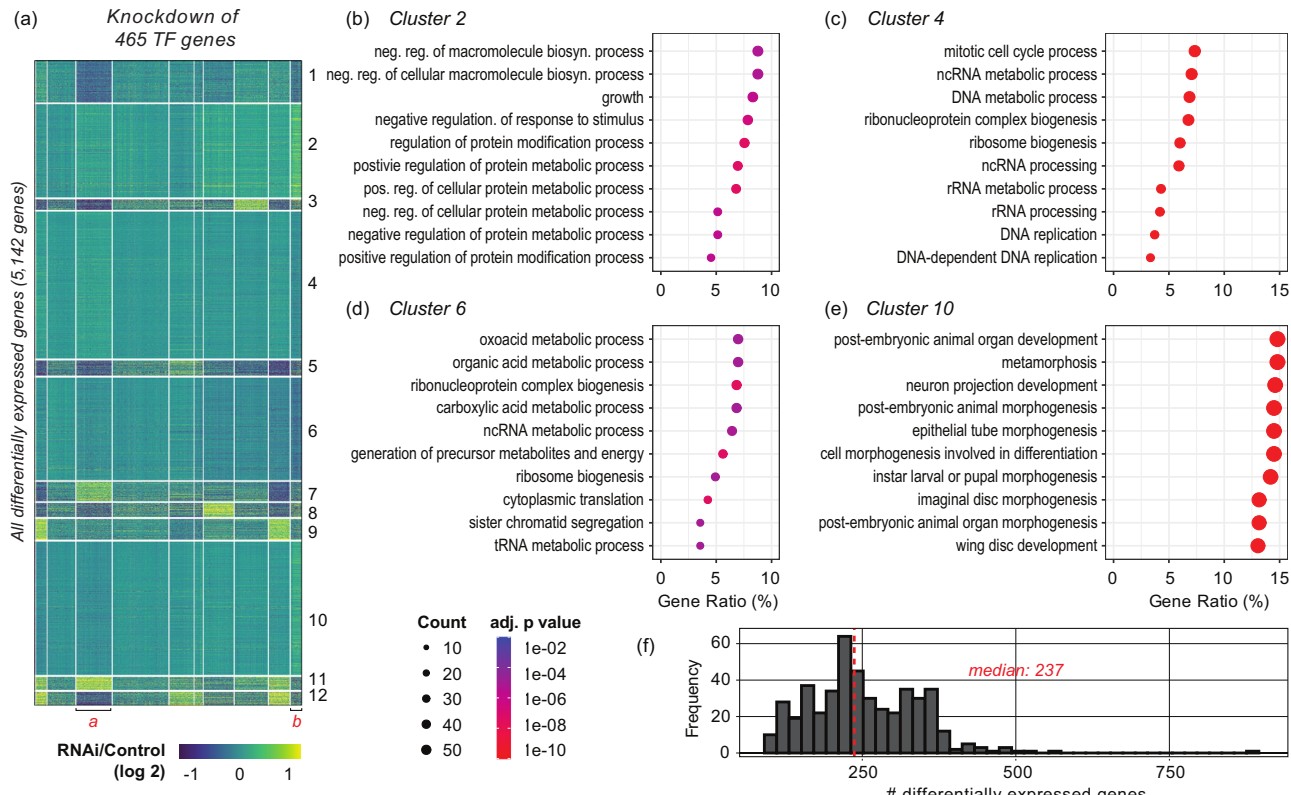

**Fig. 3 Gene expression changes upon targeted knockdown of Drosophila TFs in S2 cells. a** A heatmap summarizes differential gene expression between Drosophila S2 cells treated with TF RNAi versus control RNAi (dsRNA for *E. coli LacZ* gene). Columns. Depletion of 465 TFs. Rows. Gene expression changes in the log2 scale. Genes that are not differentially expressed from any of the TF RNAi experiments were not shown in this heatmap. The results are clustered using the k-means clustering method. **b–e** Results from Gene Ontology (GO) analysis for the clustered genes in (**a**). The four clusters with the largest gene sets are demonstrated. Circle size. A number of genes are associated with a specific GO term or hits. Circle color. Adjusted *p* value. **f** A histogram that summarizes numbers of differentially expressed genes. Vertical red line. Median of the observation.

target TFs using the bathing method[45]. For the most of targets (452 out of 488), we used two different long dsRNAs and these were biologically duplicated to give four measurements per gene. Because we were interested in using these data to trace gene network perturbations without confounding effects from long-term RNAi treatment, we chose the 1-day incubation for the main experiment. Overall, we observed that the median RNA expression was 23.4% after the knockdown (see Methods).

These knockdowns disrupted coherent pathways, not only those controlling the cell cycle, but also cellular differentiation and developmental processes (Fig. 3). We observed subsets of genes with specific functional categories display correlated gene expression changes upon the depletion of different TFs (Fig. 3b–e). For example, the genes in Cluster 2 in Fig. 3a have enriched Gene Ontology (GO) terms in "growth" (GO:0040007, adj. *p* value = 1.89e–06, Hypergeometric test corrected with

Benjamini and Hochberg method) and "regulation of protein modification process" (GO:0031399, adj. *p* value = 3.86e–08, Fig. 3b). On the other hand, the genes in Cluster 4 play a role in cell proliferation, which should be the most important phenotypic characteristic of dividing tissue-culture cells. Such genes showed enrichment of GO terms in "mitotic cell cycle process" (GO:1903047, adj. *p* value = 3.47e–12), "DNA replication" (GO:0006260, adj. *p* value = 4.21e–12) and "ribosome biogenesis" (GO:0042254, adj. *p* value = 8.76e–15) that are required for cell cycle progression (Fig. 3). Cluster 6 genes also function in cell proliferation with "ribonucleoprotein complex biogenesis" (GO:0022613, adj. *p* value = 3.25e–08) and "sister chromatid segregation" (GO:0000819, adj. *p* value = 1.29e–05), although this cluster includes metabolic genes as well (Fig. 3). Remarkably, RNAi-based differential expression of Cluster 4 and 6 genes demonstrated an anti-correlation to that of Cluster 10

genes (Pearson's $r = -0.86$), which comprise developmental genes that function in "post-embryonic animal organ development" (GO:0048569, adj. $p$ value $= 2.30e–44$), "metamorphosis" (GO:0007552, adj. $p$ value $= 4.65e–44$), "neuron projection development" (GO:0031175, adj. $p$ value $= 2.57e–42$), or "wing disc development" (GO:0035220, adj. $p = 4.59e–42$, Fig. 3).

We further noticed that knockdowns of TFs with overlapped functions result in positively correlated gene expression changes. For example, we did GO enrichment analysis for TFs in the group "a" in Fig 3a. We found that TFs in group "a" are enriched in "instar larval or pupal morphogenesis" (GO:0048707, adj. $p = 2.05e–04$) and "metamorphosis" (GO:0007552, adj. $p = 2.93e–04$). Intriguingly, those TFs, when depleted, commonly reduced metabolic gene expression that function in the "amino sugar catabolic process" (GO:0046348), which is the function of genes in Cluster 8 (genes in Cluster 8 in Fig. 3a are enriched in GO:0046348 with adj. $p = 0.00011$) or "monocarboxylic acid catabolic process" (GO:0072329), which is the function of genes in Cluster 12 (genes in Cluster 12 in Fig. 3a are enriched in GO:0072329 with adj. $p = 0.012$). Likewise, we found a group of TFs ("b" in Fig. 3a) whose knockdown upregulates growth or protein metabolism-related genes (Cluster 2 in Fig. 3a), but downregulates cell cycle or carbon metabolism genes (Cluster 6 in Fig. 3a). Altogether, these observations indicate that coordinated responses exist in the *Drosophila* S2 gene network against the perturbations from individual TF knockdown. We then utilized this RNAi data to build an RNAi-based network and used this network as the benchmark dataset for performance evaluation of the competing methods. For the downstream analysis, we defined differentially expressed genes based on log 2-fold changes (either $>1$ or $<-1$) or adjusted $p$ value ($<0.05$). Details of the RNAi network and its overlap with the other prior networks are illustrated in Table 2.

Using the S2 cell gold standard from the RNAi experiments, we evaluated the performance of our NetREX-CF with MerlinP[7], NetREX[2], LassoStARS[6], the original CF[18], PriorSum, GRNBoost2[43], GENIE3[42], and the method that makes a random choice. We describe the parameter selection for competing methods in Supplementary Note 3.

Figure 4a exhibits the comparison between the competing algorithms in terms of average rankings of TFs for each target gene in the benchmark dataset. The curve of NetREX-CF is below all other methods, indicating, on average, NetREX-CF ranks TFs identified from RNAi knockdown data lower than other competing methods. In addition, Fig. 4b illustrates the comparison between the competing algorithms in terms of average rankings of target genes for each TF in the benchmark dataset. Intriguingly, many of the tested methods failed to perform better than the random method. Similar observations have been made by previous studies[3,4]. Only NetREX-CF displayed better performance than the random methods, in terms of the overall ARS, far surpassing all the other methods (Fig. 4c). However, when the ranks of TFs were considered, NetREX-CF has better performance for 182 out of 458 TFs; its prediction power started to be weaker than the random method for the rest of the 276 TFs, analogous to other methods. Still, this result is far better than other methods that only predicted approximately the top 100 TFs (GENIE3 and GRNBoost2), or performed worse than the random methods for all the TFs. We believe that this is due to the TFs that regulate a large number of target genes (see Discussion).

For further comparison between the benchmark data and the GRNs inferred by each method, we performed Gene Set Enrichment Analysis (GSEA). We tested whether the target genes for each TF in the benchmark dataset are statistically enriched on the top of the ranked gene list ($p$ value $< 0.01$), generated by each method. We found that NetREX-CF achieves

the best performance, compared to the other methods, by significantly recovering behaviors of 94 TFs in the benchmark network (Fig. 4d). It is worth mentioning that this number is larger than the summed number of TFs confirmed by the original NetREX ($=38$) and CF ($=50$), which represents the synergy between the two algorithms in NetREX-CF. GENIE3 followed NetREX-CF with 76 TFs recaptured.

To evaluate the biological relevance of the inferred GRNs, we compared NetREX-CF and GENIE3, which are the two best methods based on our validation. We performed GSEA analysis and investigated TFs whose inferred target genes are significantly enriched for two Gene Ontology (GO) terms: mitotic cell cycle (GO:0000278), or nervous system development (GO:0007399). Potential target genes under these functional terms are expected to be activated or repressed, respectively, in *Drosophila* S2 cells, which is a tissue-culture cell line derived from the late embryos[51]. NetREX-CF predicted a total of 172 TFs for the regulation of mitotic cell cycle genes when the number was 26 for GENIE3 (Fig. 4e). There was a total of 10 TFs inferred by both methods. The overlapped TFs include Myb, which is a fly ortholog of human MYB and MYBL1, as well as Sin3A. These TFs are known to regulate the G2 or G2/M transition of the cell cycle[52,53]. We also found that Grp, a product of *grp* (*grapes*) gene and the fly ortholog of human *CHEK1*, from the list. The TF plays a role in cell cycle checkpoint in response to DNA damage[54]. In addition, while the overlap was partial, both methods identified members of the *E2F/RB* transcription factors, or *Myb*-interacting proteins, which include *E2f2* and *Rbf2* (GENIE3), *E2f1*, *Rbf*, *Mip120*, and *Mip130* (NetREX-CF). These TFs form a complex and regulate the G1/S transition of the cell cycle[55,56]. Therefore, both NetREX-CF and GENIE3 could efficiently predict TFs that regulate mitotic cell cycle genes.

Since NetREX-CF (172) could predict more TFs than GENIE3 (26) as the regulator of the same functional group (mitotic cell cycle), we asked whether NetREX-CF has better efficacy than GENIE3. Remarkably, only NetREX-CF identified a set of TFs that are related to Polycomb and Trithorax Group [Polycomb (Pc), Polycomblike (Pcl), Pleiohomeotic-like (Phol), Enhancer of zeste E(z), Additional sex combs (Asx), and Scm-related gene containing four MBT domains (Sfmbt), Trithorax-related (Trr)]. While these TFs are well known for their regulation of development and differentiation[57], there also are involved in cell cycle control[58–62]. Moreover, only NetREX-CF, but not GENIE3, predicted members of the mediator complex (*CycC*, *MED1*, *MED23*, *MED24*, and *MED26*) as regulators of the mitotic cell cycle genes. Multiple studies have reported the role of these components in cell cycle control or proliferation[62–65]. We obtained a consistent result for the TFs that control the nervous system development (Fig. 4f). Both NetREX-CF and GENIE3 identified key regulators, such as *Kay* (Kayak, fly ortholog of human FOS) and *Jra* (Jun-related, fly ortholog of human *JUN* and *JUND*), or *Foxo* (Forkhead box, sub-group O), which are well studied for their roles in nervous system development in fruit flies and/or human[66–70]. However, only NetREX-CF could capture, for example, *Sox14*, which is one of the master regulators of neurogenesis in *Drosophila*[71]. Collectively, we observed that NetREX-CF outperforms GENIE3 in inferring biologically relevant GRNs for those functional categories.

Importantly, NetREX-CF surpassed the competing methods even when we did not use the TF binding prior (e.g., ChIP-Seq). ChIP experiments largely rely on the availability of suitable antibodies that can pull down TFs under the cross-linked condition. Therefore, we tested how the absence of ChIP data would change the performance of NetREX-CF using our *Drosophila* S2 cell benchmark dataset (Fig. 4g–i). The performance of GRNBoost2 and GENIE3 is not altered since they do

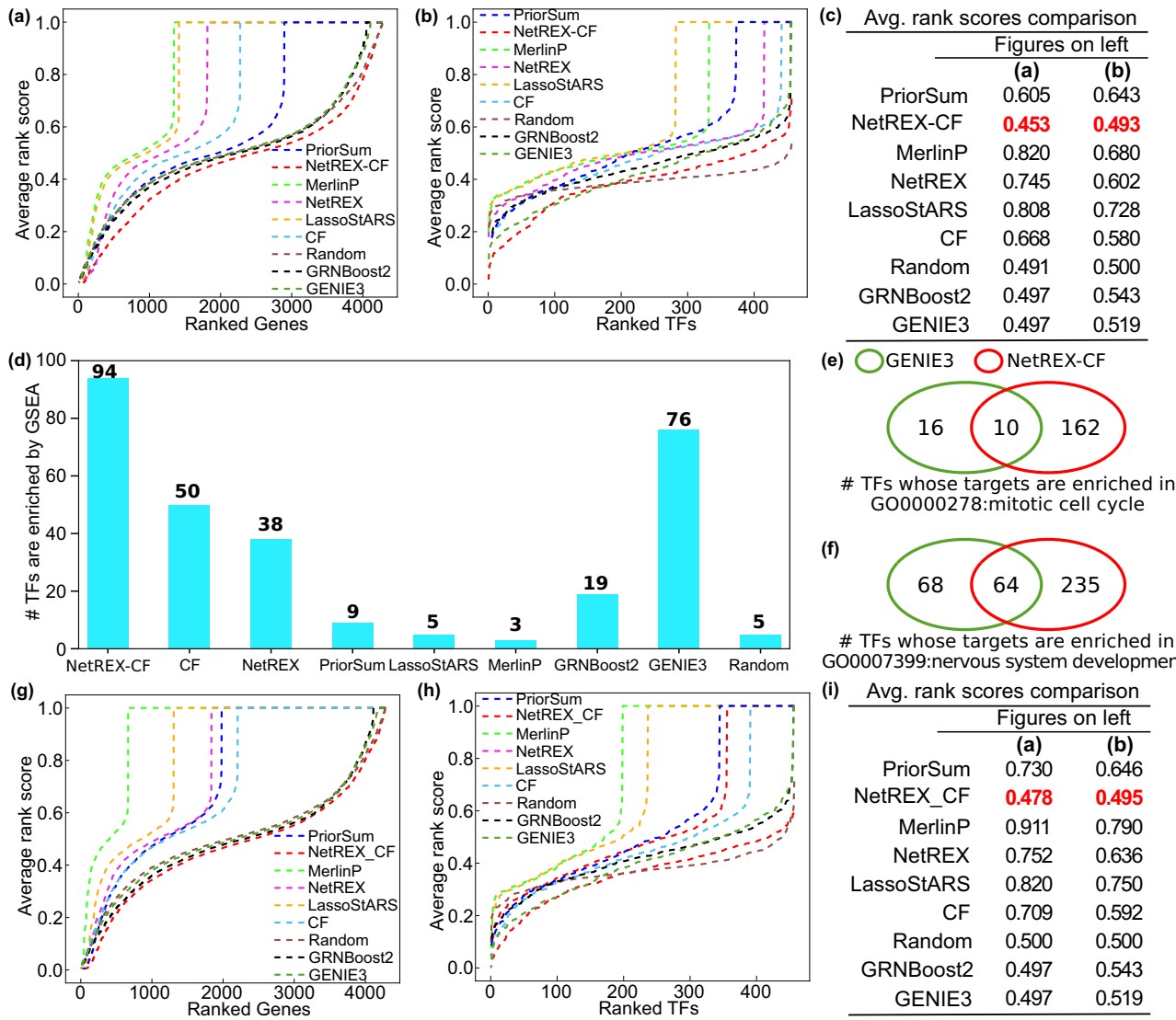

**Fig. 4 Performance comparison for all competing algorithms on the S2 cell line. a** The performance of the methods on the task of predicting regulating TFs: for each algorithm, we compute for each gene the rankings of edges in the benchmark dataset adjacent to it, sort them in descending order, and plot the sorted average rankings. **b** The performance of the methods on the task of predicting regulated genes using a measure similar to in (**a**) but focusing on genes regulated by TFs. **c** Summary of the average rank scores in (**a**) and (**b**). The lowest scores are highlighted in bold red. **d** The bar plot indicates the number TFs whose target genes in the benchmark dataset are significantly enriched at the top of the ranked gene list ($p$ value < 0.01 with 1000 random permutations), as predicted by each method in the GSEA analysis. **e, f** The Venn diagram between TFs is identified by the top two methods in (**b**): GENIE3 and NetREX-CF. The target genes of TFs shown in are enriched in the GO term of the mitotic cell cycle (**e**) and nervous system development (**f**), respectively. **g–i** The same plots as (**a**) and (**b**). The only difference is that all the competing methods do not use the ChIP prior.

not use any prior information (ARS = 0.497 for the TF predictions for both). In general, the performance of the prior-based methods (e.g., MerlinP, LassoStARS, the original NetREX, and PriorSum) became worse (compare Figs. 4a, g) when no TF binding prior was used (compare Figs. 4a, g). This was the case for NetREX as well (ARS 0.453 vs. 0.478 with, and without, the ChIP results, respectively), underscoring that the use of the TF binding prior greatly improves the GRN inference. However, NetREX-CF still outperforms all the competing methods even without the information, although the difference with the second-best, GENIE3, becomes smaller (ARS 0.478 vs. 0.497 for the gene prediction). This finding shows that NetREX-CF still functions and outperforms alternative approaches in the absence of ChIP-Seq or ChIP-chip data. However, we advise employing the TF binding prior wherever possible to get the best prediction with NetREX-CF.

**Reconstruction of cell-type-specific GRNs from human single-cell RNA-Seq studies**. ScRNA-Seq technology is rapidly adopted by biologists for uncovering cell-type-specific transcription programs. To demonstrate that NetREX-CF can generate cell-type-specific GRNs using scRNA-Seq results, we benchmarked all competing methods using previously published human studies. To build GRN for human hepatocyte-like cells (hHep GRN), we collected gene expression profiles from a scRNA-seq analysis of induced pluripotent stem cells (iPSCs) in two-dimensional culture differentiating to hepatocyte-like cells[31]. We utilized the following datasets as priors: (i) functional prior GRN from STRING[72,73], and (ii) non-specific prior GRN from DoROthEA, RegNetwork, and TRRUST[73–76]. To evaluate the inferred hHEP GRN, we extracted all the cell-type-specific GRNs from ENCODE, ChIP-Atlas, and ESCAPE databases for ChIP-Seq data from the same or similar cell type[73,77–79]. The summary of the

**Table 3 Statistics for the scRNA-Seq data for hHEP and hSCE.**

|  | No. of TFs (expressed 500 cells) | No. of genes (expressed 500 cells) | No. of cells (with 3 gene expressed) |
|---|---|---|---|
| hHEP | 645 | 6061 | 425 |
| hESC | 745 | 7221 | 758 |

**Table 4 Statistics of the prior and benchmark networks for hHEP and hESC.**

|  |  | No. of TFs | No. of genes | No. of edges | No. of overlap with STRING | No. of overlap with non-specific | No. of overlap with cell-specific |
|---|---|---|---|---|---|---|---|
| hHEP | STRING | 620 | 5611 | 35,743 | 35,743 | 2806 | 935 |
|  | Non-specific | 566 | 3258 | 26,579 | 2806 | 26,579 | 9493 |
|  | Cell-specific (benchmark dataset) | 40 | 5692 | 87,929 | 935 | 9493 | 87,929 |
| hESC | STRING | 705 | 3830 | 41,331 | 41,331 | 3456 | 1381 |
|  | Non-specific | 652 | 6621 | 32,300 | 3456 | 32,300 | 9880 |
|  | Cell-specific (benchmark dataset) | 55 | 6542 | 112,590 | 1381 | 9880 | 112,590 |

single-cell RNA-Seq, as well as the prior and benchmark GRNs, are elaborated in Tables 3 and 4, respectively.

We compared our NetREX-CF with the prior-based methods (NetREX, MerlinP, and LassoStARS), which can incorporate the prior information to build the hHEP GRN as well as the expression-based methods (GRNBoost2 and GENIE3), which rely on only the scRNA-Seq data to build the hHEP GRN. These two methods were the best-performing methods for predicting directed networks using scRNA-Seq data in a recent evaluation, and therefore, provide a good baseline for the comparison[4]. NetREX-CF achieves the lowest overall ARS by a large margin (Fig. 5a, ARS = 0.349, compared to 0.5 from the random method), indicating that the rankings of the TFs predicted by NetREX-CF are much lower than the competing methods from the benchmark. In the case of their target gene prediction, we found that all methods have comparable degrees of performance [Fig. 5b, c, ARS 0.486–0.574 with NetREX-CF performing the best]. We think that this is because the TFs in the benchmark dataset control thousands of genes, resulting in an ARS of about 0.5 (see the discussion section).

As the second example, we built the human embryonic stem cell GRN (hESC GRN) by using scRNA-Seq results of the differentiation protocol to produce definitive endoderm cells from human embryonic stem cells[32]. Analogous to what we performed for building hHEP GRN, we utilized a functional prior network based on the STRING DB, non-specific prior networks from other databases, and cell-type-specific GRN from ChIP-Seq datasets (Tables 3 and 4 for the summary information). Consistent with our observation in the hHEP GRN, NetREX-CF demonstrated superior performance to the other methods in recovering the rank of TFs in the benchmark dataset (Fig. 5d, ARS = 0.366 compared to 0.500 from the second-best method, GENIE3). However, in agreement with the hHEP GRN results, recovering the rank of target genes in the benchmark dataset displayed less distinguishable performances for all the methods (Fig. 5e, f, ARS 0.489 (best, NetREX-CF) to 0.575 (worst, PriorSum)).

## Discussion
Data integration and predictive modeling are the two key tasks of Computational Biology. However, these two tasks are rarely considered together. GRN reconstruction is an example of an important and challenging computational biological problem that

can benefit from both approaches. Here we propose a method that combines a machine learning-based data integration strategy (CF) and a gene expression modeling approach (NCA) into one global iterative optimization strategy where the machine learning component informs the expression-based modeling component and vice versa. Our integrative GRN reconstruction method, NetREX-CF, can infer cell-type-specific GRNs by maximizing different types of available information, including non-specific priors. NetREX-CF outperforms previous computational methods for this task demonstrating the power of our integrative approach.

Our contribution includes the introduction of new benchmark datasets that can be widely used for GRN inference. The yeast datasets, including YEASTRACT, have been extensively used as the testbed by the community. However, as a unicellular eukaryote, yeast does not have many TFs [such as Polycomb Group (PcG) Proteins] that are essential for multicellular development. In addition, a convincing evaluation would not result from employing the yeast dataset as the exclusive source of assessment basis. In this study, we offered a different set of the benchmark dataset based on a large-scale RNAi treatment of *Drosophila* S2 cells to get beyond the current restriction on utilizing just the yeast data. It is important to note that this dataset was created specifically for GRN inference experiments. For example, indirect effects are unavoidable for any kind of loss-of-function study, including RNAi, CRISPR, and even drug treatments. Therefore, we have incubated our cells with RNAi reagent for only one day to minimize the prolonged indirect effect. We achieved an excellent knockdown efficiency at the RNA level (Supplementary Fig. 1); thus, the target TF is expected to be reduced by more than 50% at the protein level, considering the doubling time of *Drosophila* S2 cells (~1 day[80]). We also carefully evaluated potential off-target effects that the RNAi reagents might cause, which were negligible (Supplementary Fig. 1e). We strongly believe that this will be one of our community's best datasets for the study of GRN inference, which can supplement the yeast gold standard.

In our experiments for building S2 cell GRN, we observe that the random method outperforms most of the competing methods. Many previous works[3,4] have the same observation that the random method performs better than most of the methods on GRN inference. In the DREAM5 network inference challenge[3], all the state-of-the-art GRN inference methods perform similar or even worse than the random method for building the yeast GRN. The majority of the state-of-the-art GRN inference approaches do not outperform random methods on several measures for

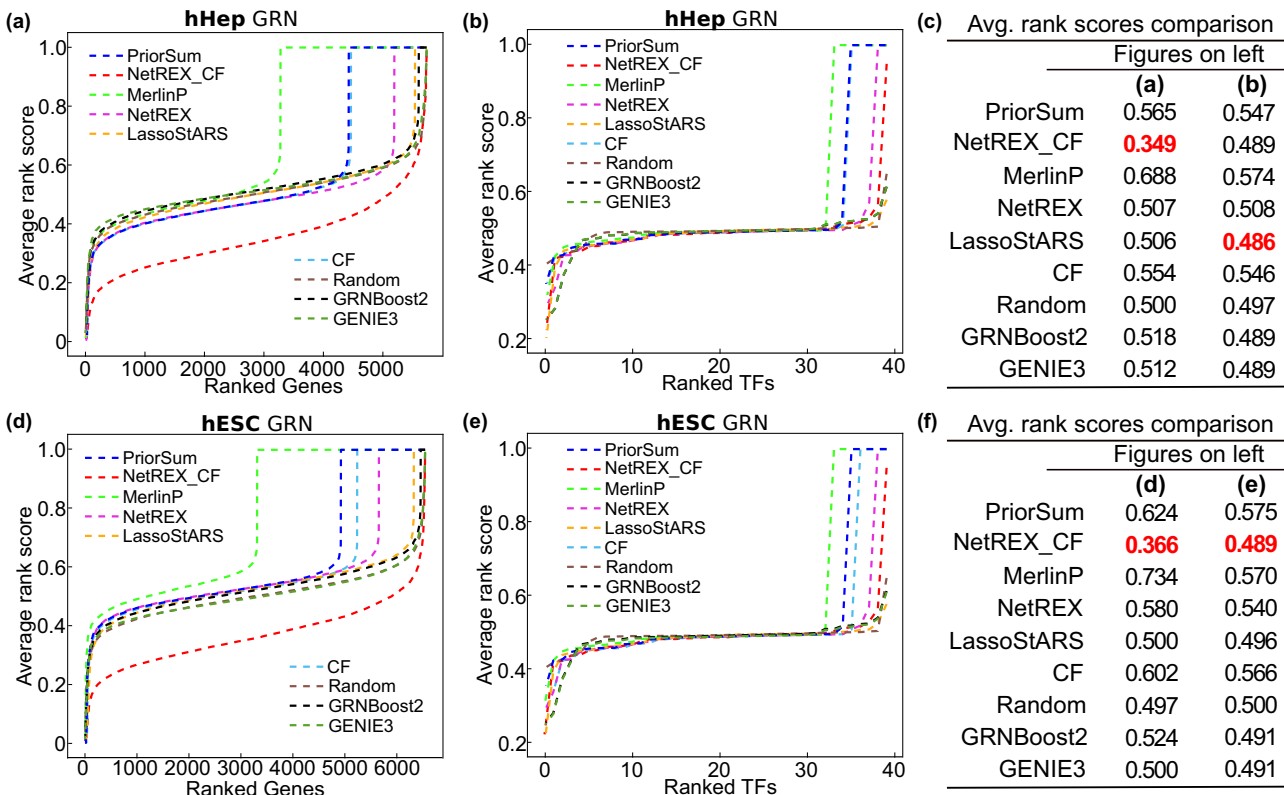

**Fig. 5 Performance comparison for all competing algorithms on building cell-specific GRNs from scRNA-Seq results. a** The performance of the methods on the task of predicting regulating TFs in hHep GRN. **b** The performance of the methods on the task of predicting target genes in hHep GRN. **c** Comparison of the overall ARS in (**a**) and (**b**). The lowest scores are highlighted in bold red. **d** The performance of the methods on the task of predicting regulating TFs in hESC GRN. **e** The performance of the methods on the task of predicting target genes in hESC GRN. **f** Comparison of the overall ARS in (**d**) and (**e**).

developing particular cell-specific GRNs, according to recent benchmarking work for single-cell GRN inference[4]. All of these show that creating GRNs is still a difficult computational endeavor that requires further work.

In this work, we select the Average Rank Score (ARS) as the metric to assess the GRN inference techniques rather than the Area Under the Precision-Recall (AUPR). This is so that we can assess whether the TF or target gene rankings in the benchmark dataset have been recovered. The ARS is a widely used metric[18], and focuses on the rankings of the edges in the benchmark network. In contrast, when computing the AUPR score, we treat the edges in the benchmark data as true-positives but the edges not in the gold standard as true-negatives, which might be misleading when the benchmark dataset is incomplete. Therefore, we used ARS to evaluate the performance of GRN inference based on the rankings of the gold standard. In addition, we notice that when a TF has thousands of target genes (for example, Myc), ARS tends to be around 0.5. However, we would like to underscore that this does not create any problem for the comparisons of GRN inference methods. It is because the only way to achieve a low ARS is where the ranking of the TFs/genes in the benchmark dataset is low; the condition is fair for all the GRN inference methods that we have tested.

To the best of our knowledge, most previous works use the fraction of the correctly predicted TF-gene pairs to show the competitiveness of their method. However, such a measure does not assess which method better predicts regulators for a given gene. In this study, we propose a new rank-based metric that is gene-centric. Using the new metric, we can evaluate whether the predicted regulators for a gene are consistent with the benchmarking data. As can be appreciated from Fig. 2a, this is a much

more challenging task, and our method outperformed competing approaches by the new metrics. This gives us confidence that our method is very competitive in predicting regulators in general. From our experiments for scRNA-Seq, we demonstrated that we could pinpoint predicted cell type-specific regulators based on cell type-specific gene expression. However, we have no objective test to judge whether NetREX-CF is more successful than others in identifying "important" regulators.

We believe that the general approach presented in this study together with collected data provides not only an important step toward reconstructing better GRNs, but it has also the potential to become a paradigm for addressing other optimization problems in computational biology.

## Methods
### The experimental procedure of the RNAi experimental data
*Selection of TFs for RNAi-mediated knockdown and setting up for RNAi treatment.* We aimed to knock down every TFs expressed in *Drosophila* S2 cells for work in gene expression network reconstruction. To facilitate gene selection, we used a curated list of known and predicted transcription factors based on experimental evidence of DNA binding and homology for example provided by the Harvard *Drosophila* RNAi Screen Center (DRSC, Boston, MA). The list also includes genes with potential transcription cofactor activities and chromatin association with ranks (Supplementary Data 1). We then removed TFs that were not expressed in S2R+ cells (an isolate of Drosophila S2 cells, used in our RNAi treatment and RNA-Seq) according to our previous work[23] and selected the top 488 ranked TFs (Ranks 1–7 in Supplementary Data 1) as RNAi targets. We used two dsRNA reagents per TF encoding gene whenever possible (452 targets), although in some cases only one reagent was used (36 targets). The RNAi treatments were duplicated and performed on a total of twenty 96-well plates (1920 samples, Supplementary Fig. 1). On each 96-well plate, we had two control RNAi samples. First, cells treated with dsRNA for Diap1 (Death-associated inhibitor of apoptosis 1 = thread) served as a visual control of dsRNA delivery since its function is essential for apoptosis prevention[81]; the knockdown leads to noticeable cell death. Second, each plate

included cells treated with dsRNA for *E. coli LacZ* gene as a sham control (hereafter, *dsLacZ*). We used these *dsLacZ* treated cells from each plate as a control for our differential expression analyses, which minimized the potential batch effects.

*S2R+ cell culture and treatment of dsRNA.* *Drosophila* S2R+ cell culture and RNAi-mediated knockdown of TFs were performed at the Harvard *Drosophila* RNAi Screen Center (DRSC, Boston, MA). 2.7 ug of dsRNA in 27 uL RNase-Free water was pre-loaded onto 96-well plates before the cells. Cells were diluted from confluency by spinning at 1300 rpm for 3 min and adding serum-free media to have 1e5 cells/mL. We transferred 53 uL of the suspension (~5300 cells) to the dsRNA-containing 96-well plates. We incubated cells for 30 min to settle down, then added 160 uL of complete media with 10% FBS per well. We incubated the cells at 25 °C for 24 h. We removed media using a 12-well vacuuming wand. We lysed the cells with 150 uL of RLT buffer (Qiagen, Valencia, CA, Cat. No. 79216) by pipetting up and down five times. Lysates were stored at −80 °C until RNA extraction. Detailed protocols, the list of transcription factors, and raw and processed expression profiles can be found at Gene Expression Omnibus (https://www.ncbi.nlm.nih.gov/geo/) with accession ID GSE81221. Each plate had two control wells. We added dsRNA for LacZ in one well, and used it in our differential expression analysis. In the other well, we treated cells with dsRNA for DIAP1; we observed cell deaths from the wells for our visual inspection of dsRNA uptake. We used two different dsRNA reagents for 452 target genes; only one reagent was used for the rest 36 targets.

*Preparation of RNA-Seq library.* We extracted total RNA using RNeasy 96 kits (Qiagen, Valencia, CA) based on the manufacturer's guideline (Protocol for Isolation of Total RNA from Animal Cells using spin technology, Cat#19504). We prepared PolyA+ RNA-Seq libraries as in described previously[82]. We added spike-in RNA from the External RNA Control Consortium (ERCC, Baker et al.)[83] during the RNA fragmentation step. For RNAi control samples (dsRNA for *LacZ*), we added ERCC pool 78A. For the rest of the samples, we added ERCC pool 78B. They are from the National Institute of Standards and Technology (Standard Reference Material 2374[84]). We incorporate dUTP during the second strand synthesis of the reverse transcription step in order to generate strand-specific libraries. We ligated 24 different indexed adapters from TruSeq v2 kit (Illumina, San Diego, CA) to reverse-transcribed cDNA for multiplexed sequencing. We enzymatically removed dUTP-incorporated strands from the ligation outcome before PCR using Uracil-DNA glycosylase (Thermo Fisher Scientific, Waltham, MA). Sequencing was done with Illumina HiSeq 2500 (Illumina, San Diego, CA) as 50 bp single-end sequencing at the NIDDK Genomics Core (Bethesda, MD).

*Data processing of RNA-Seq.* In order to evaluate replicate reproducibility, we compared gene expression levels in log2 FPKMs. Correlation between replicates ranged from 0.99 to 1.0 (Pearson's r, Supplementary Fig. 1b). When all samples were pairwisely compared, Pearson's r coefficients were between 0.92 and 1.0. From the pairwise comparison, we observed a small degree of batch effects. We performed our experiments in a standard 96-well plate format, where we treated S2R+ cells with dsRNA, isolated cellular RNA, and prepared libraries in 96-well plates. When we compared gene expression levels from different plates (Supplementary Fig. 1b), we observed a lower correlation among plates especially when Plates 1–8 were compared to Plates 9–16, although the difference was very small (plate numbers are by our definitions). We do not know if the observed batch effect is due to biological reasons (e.g., confluency, or nutritional condition of cells that were plated), technical reasons, or both. To minimize the impact of this batch effect, we performed our differential expression analysis by comparing dsRNA-treated samples to the controls (*dsLacZ*) from the sample plates.

Before the differential expression analysis, we wanted to estimate the knockdown efficiencies for the selection of "better" RNAi samples, but we found that dsRNA reagents are still detected after the rigorous polyA+ mRNA enrichment step (Supplementary Fig. 1c, shown as "double-stranded" in our strand-specific RNA-Seq). To account for reads from residual dsRNA reagents in differential expression analysis, we removed any reads that are entirely mapped to dsRNA target regions from both target-specific RNAi samples as well as controls (dsRNA for LacZ). In doing that, we used featureCounts as above using a custom gene annotation file that includes only dsRNA reagents but counted for both strands. Then we performed differential expression analysis using an R package DESeq2 version 1.30.0[85] to determine gene expression changes as well as the target gene deletion. We filtered out the outcome based on the expression cutoff above (FPKM = 0.84). The calculated RNAi efficiency is provided in Supplementary Data 1. We observed that 22% of the dsRNA led to less than 50% of the target gene expression reduction (Supplementary Fig. 1d). For our network construction, therefore, we used RNAi samples with better efficiency when two different reagents were used for a single target. The median knockdown efficiency was 76.6% reduction at the RNA level.

**Collecting S2 cell expression profiles**. We identified 600 S2 cell RNA-Seq datasets in the SRA (GSE117217). We downloaded and extracted FASTQs using fastq-dump (v2.10.1; –split-3 –skip-technical –minReadLen 20) and trimmed reads for low quality bases and adapter sequence using atropos (v1.1.25[86]; -q 20). We aligned reads using hisat2 (v2.1.0[87]; –max-intronlen 300000 –known-splice-sites),

against the *Drosophila* reference genome (Release 6 plus ISO1 MT; GCA_000001215.4), and generated read counts using featurecounts (Subread package v1.6.4[88]).

**Construction of S2 cell prior networks**. We identified 250 S2 cell TF ChIP-Seq datasets along with their input controls (Supplementary Data 2). We downloaded and extracted FASTQs using fastq-dump (v2.10.1; –split-3 –skip-technical –minReadLen 20) and trimmed reads for low quality bases and adapter sequence using cutadapt (v2.7[89]; -q 20). We aligned reads using bowtie2 (v2.3.5[90]) against the *Drosophila* reference genome (Release 6 plus ISO1 MT; GCA_000001215.4) and filtered low quality alignments and multi-mapping reads with samtools (v1.9[91]; -q 20). Next, we merged technical replicates and removed duplicated reads with picard MarkDuplicates (v2.21.6[92]). To map putative TF binding sites, we used macs2 (v2.2.4[93]; -g dm –bdg –nomodel –extsize 147) to call peaks. We constructed the ChIP-Seq weight matrix similar to the motif matrix. First, we removed peaks that overlapped multiple genes. Next we assigned a peak to a gene if it overlapped the gene body plus 1kb upstream. Finally, we created a TF-Target gene weight by calculating the average peak score.

We downloaded TF DNA binding motifs from 5 *Drosophila* databases (OnTheFly[46], FlyFactorSurvey[47], FLYREG v2[48], iDMMPMM[94], and DMMPMM[50]). We mapped motifs to the *Drosophila* reference genome (Release 6 plus ISO1 MT; GCA_000001215.4) using FIMO (MEME Suite v5.1.0[95]). Next, we wanted to assign binding locations to individual genes. First, we filtered out binding locations that overlapped 2 or more annotated genes (FlyBase r6.26). Next, we assigned all locations that mapped within the gene body plus 1 kb upstream of the gene start. Finally, we combined motif binding sites from all five databases and used the maximum binding score for a given TF-Target Gene pair to build the motif weight matrix.

To increase the number of TFs with prior information, we used 200 S2 cell expression profiles to construct a co-expression network containing all 457 TFs. These 200 samples were then excluded from GRN prediction. To build the co-expression network, we first calculated Pearson product-moment correlation coefficients between TFs and all genes. We then removed edges between TF-gene pairs if their correlation coefficient is <0.7.

**Defining the gold-standard edges based on the RNAi experiment**. We generated a gene-level read count matrix. We used the gene annotation from Flybase version 6.26[96]. We performed differential expression analysis to measure gene expression changes between the samples with knockdown of transcription factors versus the control (dsRNA for LacZ). We used DESeq2 version 1.26.0[85]. In doing that, we adjusted the read counts on dsRNA target genes since we found that residual dsRNA reagent is captured in our RNA-Seq. From the differential expression analysis, we consider a TF-gene pair as a gold standard edge if the Log2 fold changes >1 or <−1 or the adjusted p-values are less than 0.05.

**NetREX-CF model**. Before describing the mathematical foundation of the NetREX-CF model, we provide a brief overview of the CF model and the sparse NCA-based network model, respectively. Next, we formally introduce the integration of these two models.

*Collaborative filtering model.* As illustrated in Supplementary Fig. 2, to reconstruct GRNs we might have access to several prior networks, each of which reflects a different perspective of the gene regulation process. Here we illustrate three prior networks: the Motif prior network, the Knockout prior network, and the ChIP prior network. In general, the prior networks are partial observations of the gene regulation process and therefore incomplete. The incompleteness of prior networks can be further demonstrated by Table 1 in the main paper, where there are only a small number of overlaps between the yeast prior networks and the gold standard GRN. Previous prior-based GRN reconstruction methods[9] typically make efforts to preserve those edges in the prior networks into the final GRN reconstruction but are unable to predict new edges to resolve the incompleteness of the prior networks.

CF, a machine learning technique, is an approach to mitigate the incompleteness of the prior networks. CF is able to make predictions based on partial observation. Given a set of prior networks $P = \{P^1, \ldots P^d\}$, the mathematical formulation of CF can be presented by

$$
\begin{aligned}
\min_{x_i, y_j} : \quad & \sum_{i,j} C_{ij} \left( B_{ij} - x_i^T y_j \right)^2 \\
s.t. \quad & \|x_i\|^2 \le 1, \; \forall i \\
& \|y_j\|^2 \le 1, \; \forall j.
\end{aligned}
\tag{5}
$$

We recall that $x_i$ and $y_j$ are hidden feature vectors for gene $i$ and TF $j$, respectively, and $B_{ij}$ is a binary number that equals to 1 when we observe the edge between gene $i$ and TF $j$ in any prior and equals to 0 otherwise. $B_{ij}$ encodes that predictions that feature vectors need to make. $C_{ij} = 1 + a\sum_k P_{ij}^k$ is the penalty for learning the edge between gene $i$ and TF $j$. Larger $C_{ij}$ implies $B_{ij} = 1$ and also encourages the dot

product $x_i^T y_j$ between gene feature vector $x_i$ and TF feature vector $y_j$ to be $x_i^T y_j = 1$. Details of the CF model are illustrated in Supplementary Fig. 2a.

After solving the optimization problem (5), we can use $x_i^T y_j$, $\forall i, j$ to predict edges that are not in the prior networks. Because of the constraints in (5), we know $x_i^T y_j \in [-1, 1]$ based on Cauchy-Schwarz inequality. $x_i^T y_j$ is close to 1 implies that the CF method recommends the edge between gene $i$ and TF $j$. However, to obtain reliable predictions, it is beneficial that the correctness of the edge recommendation is further confirmed by other methods.

*Sparse NCA-based network model.* Other than utilizing prior information, such as binding properties, we can use gene expression to help build reliable GRNs. Currently, the state-of-art methods to use gene expression for reconstructing GRNs are NCA-based approaches[2,5,6,12–16]. However, in order to use the NCA model, we need a prior network in addition to gene expression data. Given gene expression $E \in \mathbb{R}^{n \times l}$ for $n$ genes in $l$ samples and a prior network $S \in \mathbb{R}^{n \times m}$, the sparse NCA-based network model can be presented as

$$\min_{S,A} : \quad \|E - SA\|_F^2 + \lambda_A \|A\|_F^2 + \lambda_S \|S\|_F^2 + \sum_{i,j} \eta_{ij} \left\|S_{ij}\right\|_0, \tag{6}$$

where the first term is the basic NCA model[12] ($A \in \mathbb{R}^{m \times l}$ is the TF activity for $m$ TFs in $l$ samples) and the second and third terms are standard regularization terms and the last term involving $\ell_0$ norm that is able to induce sparsity of the given prior network. Therefore, solving (6) would yield a refined GRN that only retains key edges from the prior network. The details of the sparse NCA-based network model is illustrated in Supplementary Fig. 2b. Since in most cases, we do not have a prior network, we need to build a reliable prior based on multiple sources of prior information.

*Formulation of the NetREX-CF model.* Here we propose to integrate both CF model and the Sparse NCA-based network model. As we mentioned that the CF model needs a way to confirm the recommended edges and the sparse NCA-based network model needs a prior network with which to work. Therefore, it is natural to combine these two models together. The CF model can recommend a prior network for the sparse NCA-based network model, and as a reward, the sparse NCA-based network model is able to confirm the recommended edges and thus allow the CF model to predict new edges. The mathematical formulation of the NetREX-CF model is

$$\min_{S,A,x_i,y_j} : \left[\|E - SA\|_F^2 + \lambda_A \|A\|_F^2 + \lambda_S \|S\|_F^2 + \sum_{i,j} \eta_{ij} \|S_{ij}\|_0\right] + \lambda\left[\sum_{ij} \Omega_{ij}\left(\Theta_{ij} - x_i^T y_j\right)^2\right]$$
$$s.t. \quad \|x_i\|^2 \leq 1, \ \forall i$$
$$\|y_j\|^2 \leq 1, \ \forall j. \tag{7}$$

The first square bracket is the sparse NCA-based model and the second square bracket is the CF model. $\lambda$ is the balance between these two models. In the NetREX-CF model, we define $\Theta_{ij} = \|S_{ij}\|_0 \oplus B_{ij} = \|S_{ij}\|_0 + (1 - \|S_{ij}\|_0)B_{ij}$ ($\oplus$ is XOR operation) to let the CF model not only predict edges in the prior networks $B_{ij}$, but also take into account the edges confirmed by the sparse NCA-based model $S_{ij}$. Furthermore, $\Omega_{ij}$ is defined as $\Omega_{ij} = \bar{C}_{ij}\|S_{ij}\|_0 + C_{ij}(1 - \|S_{ij}\|_0)$, where $\bar{C}_{ij}$ is the user defined penalty for edges confirmed by the sparse NCA-based model $S_{ij} \neq 0$ and $C_{ij}$ is the penalty for edges not in $S$ ($S_{ij} = 0$). The details of the NetREX-CF model is illustrated in Supplementary Fig. 2c, d. The details of how to select all the user defined parameters of the NetREX-CF model are elaborated in Supplementary Note 3.

Once we put the definition of $\Omega_{ij}$ and $\Theta_{ij}$ into (7) and we put $\sum_{ij}\eta_{ij}\|S_{ij}\|_0$ into the second square bracket, we have

$$\min_{S,A,x_i,y_j} : \left[\|E - SA\|_F^2 + \lambda_A \|A\|_F^2 + \lambda_S \|S\|_F^2\right]$$
$$+ \lambda\left[\sum_{i,j} \bar{C}_{ij}\|S_{ij}\|_0 + C_{ij}\left(1 - \|S_{ij}\|_0\right)\left(\|S_{ij}\|_0 + \left(1 - \|S_{ij}\|_0\right)B_{ij} - x_i^T y_j\right)^2 + \sum_{i,j} \eta_{ij}\|S_{ij}\|_0\right]$$
$$s.t. \|x_i\|^2 \leq 1, \ \forall i$$
$$\|y_j\|^2 \leq 1, \ \forall j. \tag{8}$$

Then the function in the first square bracket is continuous and we define it as $H(S, A)$. The function in the second square bracket is lower semi-continuous (Supplementary Note 1) and we define it as $F(S, X, Y)$. Clearly, we cannot separate $\|S_{ij}\|_0$ from $x_i$ and $y_i$ and put every term involving $\|S_{ij}\|_0$ together as a separated term. To the best of our knowledge, there is no known method that is able to solve the optimization problem (8). In the following, we elaborate on the algorithm we developed to solve the NetREX-CF model.

**The NetREX-CF algorithm.** Because current methods cannot solve problem (8), where the variables that need to be optimized are coupled in the non-smooth terms, we propose a GPALM algorithm that is an extension of the PALM algorithm[22]. GPALM can be used to solve this class of optimization problem that involves inseparable $\ell_0$ norm, which is when $\ell_0$ norm cannot be separated from other optimized variables as a separated term. The format of the problem that

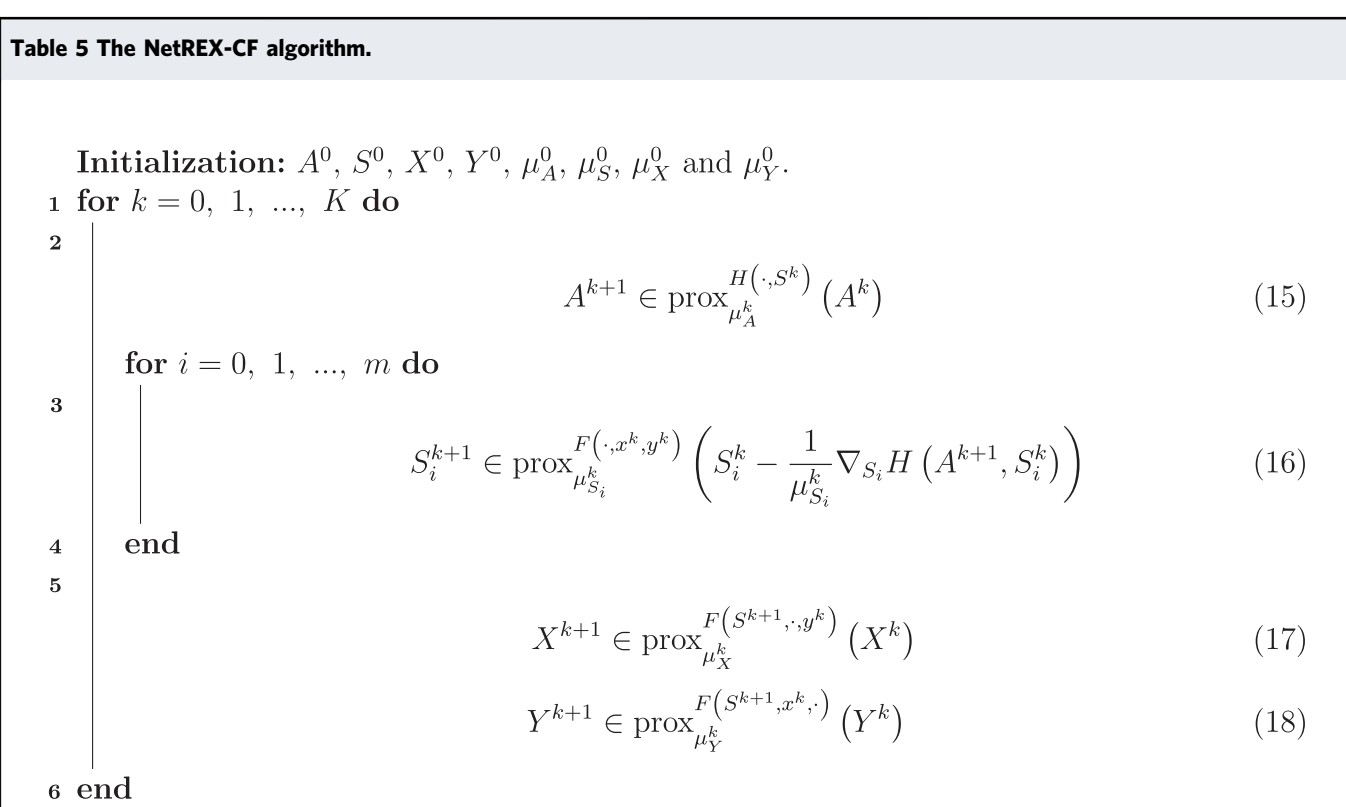

**Table 5 The NetREX-CF algorithm.**

**Initialization:** $A^0, S^0, X^0, Y^0, \mu_A^0, \mu_S^0, \mu_X^0$ and $\mu_Y^0$.

1 **for** $k = 0, 1, ..., K$ **do**

2
$$A^{k+1} \in \text{prox}_{\mu_A^k}^{H(\cdot, S^k)}\left(A^k\right) \tag{15}$$

**for** $i = 0, 1, ..., m$ **do**

3
$$S_i^{k+1} \in \text{prox}_{\mu_{S_i}^k}^{F(\cdot, x^k, y^k)}\left(S_i^k - \frac{1}{\mu_{S_i}^k}\nabla_{S_i}H\left(A^{k+1}, S_i^k\right)\right) \tag{16}$$

4 **end**

5
$$X^{k+1} \in \text{prox}_{\mu_X^k}^{F(S^{k+1}, \cdot, y^k)}\left(X^k\right) \tag{17}$$

$$Y^{k+1} \in \text{prox}_{\mu_Y^k}^{F(S^{k+1}, x^k, \cdot)}\left(Y^k\right) \tag{18}$$

6 **end**

GPALM can solve is provided in Supplementary Note 1. The GPALM algorithm and its convergence proof are provided in Supplementary Note 2. Here we directly applied the GPALM algorithm to solve our NetREX-CF model. The algorithm is listed as follows.

The proximal operator used in the algorithm is defined as:

$$\text{prox}_\lambda^\sigma(x) := \arg\min\left\{\sigma(u) + \frac{\lambda}{2}\|u - x\|, u \in \mathbb{R}^d\right\} \tag{9}$$

The proximal operator and proximal gradient methods are often applied to replace conventional smooth optimization techniques for functions that are not continuous but can be approximated by well-behaving functions (or have other nice bounding properties).

We show in the following that, for all proximal operators used in the above algorithm, we can compute the corresponding update steps by either using a closed-form that we are able to derive or by reducing the computation to a convex optimization problem.

*Update* A. The proximal operator (15) has a closed-form solution.

$$A \in \text{prox}_{\mu_A^k}^{H(\cdot,S^k)}\left(A^k\right) = \left((S^k)^T S^k + \frac{2\lambda_A + \mu_A^k}{2}I\right)^{-1}\left(E^T S^k + \frac{\mu_A^k}{2}(A^k)^T\right)^T, \tag{10}$$

where $\mu_A^k$ is the Lipschitz constant that can be computed by $\mu_A^k = \left\|(S^k)^T S^k + \lambda_A I\right\|_F$.

*Update* S. Similarly, the proximal operator (16) also has a closed-form solution.

$$S_{ij}^{k+1} \in \text{prox}_{\mu_{S_i}^k}^{F(\cdot,x^k,y^k)}\left(S_i^k - \frac{1}{\mu_{S_i}^k}\nabla_{S_i}H(A^{k+1}, S_i^k)\right) = \arg\min\left\{\left(S_{ij} - U_{ij}^k\right)^2 + c_{ij}^2\left\|S_{ij}\right\|_0\right\}, \tag{11}$$

where $c_{ij} =$

$$\sqrt{\frac{2}{\mu_{S_{ij}}^k}\left[\lambda\left[\bar{C}_{ij}\left(1 - B_{ij}\right)\left(1 + 2\left(B_{ij} - x_i^T y_j\right)\right) + \left(\bar{C}_{ij} - C_{ij}\right)\left(B_{ij} - x_i^T y_j\right)^2\right] + \eta_{ij}\right]}$$ and

$\mu_S^k$ is the Lipschitz constant that can be computed by $\mu_S^k = \left\|A^{k+1}(A^{k+1})^T + \lambda_S I\right\|_F$. Therefore, the closed solution of the above problem is

$$S_{ij}^{k+1} = \begin{cases} xU_{ij}^k, & \text{if } \left|U_{ij}^k\right| > c_{ij}; \\ \left\{0, c_{ij}\right\}, & \text{if } \left|U_{ij}^k\right| = c_{ij}; \\ 0, & o.w.. \end{cases} \tag{12}$$

*Update* X. Each row $x_i$ of $X$ needs to be updated by solving the following proximal operator.

$$x_i^{k+1} \in \text{prox}_{\mu_x^k}^{F(S^{k+1},\cdot,y^k)}\left(x_i^k\right) = \arg\min_{\|x_i\|^2 \leq 1}\left\{x_i^T \phi x_i - \varphi x_i\right\}, \tag{13}$$

where $\phi = \frac{\mu_x^k}{2}I_{h \times h} + Y^k \tilde{A} Y^{kT}$ and $\varphi = 2\bar{S}_i \tilde{A} Y^{kT} + \mu_x^k x_i^{kT}$. $\tilde{A}$ be the diagonal matrix with the values $\bar{A}_{i1}, \bar{A}_{i2}, ..\bar{A}_{im}$ on the diagonal, where $\bar{A}_{ij} = \lambda\left(\bar{C}_{ij} + (\bar{C}_{ij} - C_{ij})\left\|S_{ij}^{k+1}\right\|_0\right)$. And $\bar{S}_i$ is defined as $\bar{S}_i = \left[\left\|S_{i1}^{k+1}\right\|_0 \oplus B_{i1}, .., \left\|S_{in}^{k+1}\right\|_0 \oplus B_{im}\right]$. Since the problem becomes a Quadratically Constrained Quadratic Program (QCQP), we leave the rest to the CVXPY python package[97,98].

*Update* Y. Each row $y_j$ of $Y$ needs to be updated by solving the following proximal operator.

$$y_j^{k+1} \in \text{prox}_{\mu_Y^k}^{F(S^{k+1},x^k,\cdot)}\left(y_j^k\right) = \arg\min_{\|y_j\|^2 \leq 1}\left\{y_j^T \phi y_j - \varphi y_j\right\}, \tag{14}$$

where $\phi = X^{k+1}\tilde{A}X^{k+1T} + \frac{\mu_y^k}{2}I_{p \times p}$ and $\varphi = 2\bar{S}_j^T \tilde{A} X^{k+1T} + \mu_y^k y_j^{kT}$. $\tilde{A}$ that is also a diagonal matrix with the values $\bar{A}_{1j}, \bar{A}_{2j}, ..\bar{A}_{mj}$ on the diagonal and $\bar{S}_j = \left[\left\|S_{1j}^{k+1}\right\|_0 \oplus B_{1j}, .., \left\|S_{nj}^{k+1}\right\|_0 \oplus B_{nj}\right]^T$. Since the problem also becomes a QCQP, we leave the rest to the CVXPY python package. We extend the original PALM algorithm[22] and propose the GPALM algorithm that can solve more general problems. The format of the problem that GPALM can solve is explained in Supplementary Note 1. The actual algorithm and its convergence proof are provided in Supplementary Note 2 (Table 5)..

**Statistics and reproducibility**. Differentially gene expression analysis was implemented by DEseq2[85] on S2 cell RNAi experiments with total 400 samples (adjusted $p$ value < 0.05). Gene set enrichment analysis was done by GSEAPY[99] (v0.10.5) with 1000 random permutations. Gene Ontology enrichment analyses R package clusterProfiler[100] (3.18.1).

## Data availability

The expression data used to build the yeast GRN are obtained from the Gene Expression Omnibus (GEO) with accession IDs GSE1990, GSE4654, and GSE18. The prior data used to build the yeast GRN are obtained from GSE34306(GEO), E-MTAB-311(ArrayExpress), E-MTAB-440(ArrayExpress), and GSE6273(GEO). The S2 cell RNAi datasets generated during and/or analyzed during the current study are available at GSE81221(GEO) and GSE8975(GEO). The S2 cell gene expression data are available at GSE117217 (GEO). Source data underlying Figs. 3b–f and 4d are provided at https://figshare.com/projects/NetREX-CF_figures/151416.

## Code availability

The source code of NetREX-CF is available at https://github.com/EJIUB/NetREX_CF. The Zenodo DOI is https://doi.org/10.5281/zenodo.7178603.

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

## Acknowledgements

We want to thank Alireza Fotuhi Siahpirani and Sushmita Roy for helping us run the MerlinP program. This research was supported by National Institutes of Health grants R35GM147241-01 to Y.W. and precision health intuitive at Indiana University for Y.W. This research was also supported by the Intramural Research Program of the National Library of Medicine and the National Institute of Diabetes and Digestive and Kidney Diseases at the National Institutes of Health, USA. The RNAi library of S2 cell line RNAi experiments was provided by the *Drosophila* RNAi Screening Center (NIH NIGMS R01 GM067761 and P41 GM132087) at Harvard Medical School.

## Author contributions

Y.W., H.L., and J.M.F. contributed equally. Y.W., B.O., and T.M.P. jointly supervised this work. Y.W., B.O., and T.M.P. conceived and designed the study. Y.W., H.L., and J.M.F. conducted experiments. Y.W. and H.L. wrote the manuscript. All authors read and approved the final manuscript.

## Competing interests
The authors declare no competing interests.

## Ethics approval
The hESC data used in the paper are approved by the ethics committee with IRB Approval Number: SC-2015-0010.
