## [Transparent Peer Review File · Communications Biology]

NetREX-CF integrates incomplete transcription factor data with gene expression to reconstruct gene regulatory networksResponse to Reviewers

: We would like to thank the editors and reviewers for their helpful comments, as well as the chance to submit a revised version of the manuscript, which we have revised the manuscript in accordance with all recommendations. Please find our point-by-point response and the revised manuscript. We used blue text for the reviewers' comments, and black for our responses. We hope you will find our manuscript suitable for publication.

Reviewer #1 (Remarks to the Author: Overall significance):

The authors present in the paper a new method NetREX-CF to infer genetic network. NetREX-CF combines various prior information including motif, knockout and ChIP-seq data to predict a gene-TF edge in the network using collaborative filtering (CF). The CF-recommended regulation edge is then corroborated by a sparse NCA-based method. The network edges are predicted through an iterative joint optimization of CF-recommendation and NCA modeling. The authors validated NetREX-CF using yeast network and showed it outperformed several other competing methods. Then they applied it to reconstruct the genetic network in Drosophila Schneider 2 (S2) cell line. They also knocked down all the expressed TFs in S2 cells and generated 1920 RNA-seq data sets. Using this data, they evaluated the performance of the predicted TF-target regulations using NetREX-CF and other methods. The idea of using CF to recommend edges and joint optimization with NCA analysis is interesting and the framework is well articulated in the manuscript. The manuscript falls short on several aspects and the authors should address them before publication.

: Thank you for your reviewing our manuscript. This is an excellent summary of our study.

Reviewer #1 (Remarks to the Author: Impact):

There are many methods to reconstruct genetic network and infer important regulators.

: We thank the reviewer for raising reasonable concern. We believe that having many tools for Gene Regulatory Network (GRN) inference, in turn, reflects the trend of the post-genomics era and underscores its importance. From that point of view, NetREX-CF presents unprecedented benefits compared to the previous approaches. First, NetREX-CF is designed to utilize “non-context-specific” information for the construction of “context-specific” GRNs. For example, unlike competing methods that use only gene expression profiles for generating cell-type-specific gene networks, NetREX-CF is able to incorporate other types of information from different biological contexts (e.g., TF binding from different cell types, motif studies). Therefore, NetREX-CF can take full advantage of the plethora of datasets produced during the genomics era. Second, because of the aforementioned advantage, NetREX-CF boasts superior performance compared to previous methods. Third, in the revised version of our manuscript, we demonstrated the capability of NetREX-CF in constructing GRNs from single-cell RNA-Seq

results, which will be extremely useful for unraveling GRNs at the single-cell level. We have highlighted these points in our revised manuscript.

Reviewer #1 (Remarks to the Author: Strength of the claims):

1. **(A)** The regulatory interactions inferred by NetREX-CF is an average from a collection of gene expression data in different samples rather than in particular cell state or sample. **(B)** Also, the activity of a TF is decided by its expression level but the expression level does not necessarily reflect the activity for many TFs. The authors should discuss whether these issues would limit the applications of NetREX-CF for constructing GRNs in a specific cell type or state.

Response to 1-A: The main idea of NetREX-CF is to construct a context-specific (i.e., tissue-specific and/or cell-specific) GRN by leveraging existing GRN priors. The context of interest (tissue- or cell-type-specific) is provided by a set of expression data (tissue-specific RNA-seq or single-cell RNA-seq). NetREX-CF “rewires” the prior networks by removing and adding edges to obtain a network topology that provides the best explanation for the entirety of the expression data. Therefore, as long as context-specific expression data is used by NetREX-CF, it can build tissue- or cell-specific GRNs. In order to demonstrate that NetREX-CF is able to generate context-specific GRNs, we have demonstrated its use in human single-cell RNA-Seq analyses (hHEP and hESC cells) in the revised version of our manuscript (page 17 section 2.4).

Response to 1-B: We fully concur with the reviewer’s observation that the expression level does not necessarily reflect the activity for many TFs. Therefore, unlike the traditional approaches that directly consider the gene expression of TFs as TF activities, we combined the expression of target genes and additional information from the prior networks to estimate the TF activity based on the NCA model [12-16]. There is ample evidence in the literature to indicate that TF activities estimated by the NCA model can significantly improve the reconstruction of the gene regulatory networks, which has been demonstrated by multiple groups, including us [2]. We discussed this advantage of NetREX-CF in constructing context-specific GRNs in the revised manuscript, as suggested by the reviewer (Discussion section, page 20).

2. Inference of GRNs can provide informative insights of regulatory interactions between genes. However, this study falls short of demonstrating the power of NetREX-CF in that it does not showcase what biological insights can be gained by NetREX-CF but not by the other methods. The authors only benchmarked the reconstructed GRN in S2 cells using the extensive knockdown experiments but it is unclear what insights are gained from these analyses.

: We thank the reviewer for this helpful comment. In the revised manuscript, we performed Gene Set Enrichment Analysis (GSEA) to show that NetREX-CF can indeed provide additional biological insights that GENIE3 (one of the leading methods in the field) cannot capture. Briefly, we showed that NetREX-CF not only identifies regulatory interactions that GENIE3 picks out, but also reveals additional interactions that were curated by FlyBase, but not called by GENIE3.

The expanded section for this functional analysis can be found from page 15 the second paragraph to page 16 .

3. A useful application of GRN is to uncover important regulators in a given cell type at the systems level. Can NetREX-CF outperform the existing methods on this application? This is not discussed in the manuscript.

: Thank you for this suggestion. In our revised manuscript, we have demonstrated that NetREX-CF outperforms previous methods in generating cell-type-specific GRNs from single-cell RNA-Seq results (page 17 section 2.4).

4. Are all the prior data treated equally? How to determine the prior weights for motif, knockout and ChIP-seq data? Are these parameters needed to re-set/re-train for individual applications?

: We apologize for not providing sufficient detail here. Although we started with equal weights for each type of priors, their intersections were differently treated in the subsequent steps. We elaborate on the details in the following. We encode different prior networks $\{P^1, P^2, \dots\}$ in matrix C using, $C_{ij} = 1 + \alpha \sum_k P_{ij}^k$ where we used $\alpha = 60$, suggested by [18]. C is one of the inputs for our NetREX-CF and C_{ij} encodes the prior knowledge between the i th gene and the j th TF. If more than one prior networks suggest the regulation between the i th gene and the j th TF, then C_{ij} tends to have a larger value. In our NetREX-CF formulation, larger C_{ij} would enforce to give the regulation between the i th gene and the j th TF lower ranking. We clarified this in our revised manuscript (at the end of page 5).

Reviewer #1 (Remarks to the Author: Reproducibility):

The source code is available from GitHub.

: Thank you for your review!

Reviewer #2 (Remarks to the Author: Overall significance):

Reconstructing gene regulatory networks is a challenging but very important task for a better understanding of cellular states in various biological systems. The authors had previously developed a model NetREX that integrates TF binding prior knowledge (e.g., from ChIP-Seq experiments) and gene expression data to learn the regulatory network based on the network component analysis (NCA). In this work, the authors presented a newer version of their tool named NetREX-CF where CF stands for collaborative filtering, which is commonly seen in recommender systems. The CF approach was employed here to address the missing TF binding prior. The authors also presented a solution for the optimization problem of the joint

model (CF and NCA). As TF binding prior incompleteness issue is quite common, particularly in those less-studied organisms, the proposed idea is valuable for developing Gene regulatory network reconstruction methods (especially those methods that involve the integrating of TF-gene interaction prior). Nevertheless, I still have several major concerns regarding the NetREX-CF method that the author developed. Please find below the detailed comments.

: We thank the reviewer for the helpful comments. We hope the revised manuscript will address your concerns.

Reviewer #2 (Remarks to the Author: Impact):

See the comments in overall significance section

Reviewer #2 (Remarks to the Author: Strength of the claims):

Major comments:

1. The biggest concern that I am having is the experimental validation results. I really appreciate the authors' efforts in designing and performing the experiments on *Drosophila* to validate their model predictions. **(A)** However, as shown in Figure 4 (middle panel), when predicting the target genes for top-ranked TFs, surprisingly, the random method presents the best performance. This is a bit counter-intuitive. The author did not especially explain the potential reason for such a phenomenon. **(B)** I wonder whether this is caused by the way that authors used to define the "gold standard" from the experiments (e.g., target genes were defined with $|\log_2 fc| > 1$, $p_{adj} < 0.05$. Maybe this is too stringent. If we look at table 2, the KnockDown row only has ~50k edges, which is significantly less compared to the edges defined by other TF binding priors. **(C)** The authors did show that the predicted targets for the top predicted TFs are "more" enriched in functional terms (by GSEA). Although it's making sense that the targets for the same TF should share similar functions, Fig4. (c) is still indirect evidence to support their model predictions.

Response to 1-A: We understand the reviewer's concern. While the observation is seemingly counter-intuitive, we would like to point out that many previous works have the same observation, namely, that the random method performs better than most of the methods for certain challenging tasks (Marbach et al. 2012; Pratapa et al. 2020). In the DREAM5 network inference challenge (Marbach et al. 2012), all the state-of-the-art GRN inference methods performed similarly or even worse than the random method for building the yeast GRN (Fig. 2A in Marbach et al. 2012). Likewise, many of the widely recognized methods display only marginally better, or sometimes even worse, GRN inference compared to a random method in a systematic benchmarking (Pratapa et al. 2020).

In our study, we have a similar observation when we build the *Drosophila* S2 cell GRN. While NetREX-CF demonstrated better performance than the random method, surpassing all the other methods, most of the methods were worse than the random method. It is possible that this

observation reflects the limitation of using so-called “gold standards” for the evaluation of GRN inference as Reviewer 3 suggested (Major points 3 and 4). We carefully discussed this in the revised manuscript (page 21).

Response to 1-B: In our *Drosophila* S2 cell results, the “gold standard” targets for a TF were defined as their $|\log_2\text{fc}| > 1$ or $p.\text{adj} < 0.05$ (i.e., not “and”). We believe that this condition is not stringent, especially when we do not want to include many false positive “gold standard” targets.

Response to 1-C: We apologize for the confusion. We present Figure 4c (Figure 4d in the amended version) to demonstrate whether the inferred GRN recovers TF-gene interactions in the gold standard by using GSEA. We tested whether the gold standard target genes for each TF are statistically enriched on the top of the ranked gene list generated by each method. We updated our manuscript to make this point explicit (page 15, paragraph 2).

2. As the major advantage of this new model is its ability to deal with missing TF binding priors. The authors should directly demonstrate this point using simulations. For example, the authors could simulate the “TF binding incompleteness” by randomly dropping TF binding priors (say simulate different levels of missing values). By examining the performance of the proposed method on those simulated datasets with incomplete TF binding prior, we can evaluate the robustness of the method and further confirm whether its good at dealing with missing TF binding priors, as the authors claimed.

: We are grateful for this suggestion. It is true that TF binding priors are not always applicable, especially when ChIP-Seq analyses rely on the availabilities of “ChIP-grade” antibodies. We received similar feedback from our editor, and therefore, demonstrated that NetREX-CF can infer GRNs even without any TF binding prior, and still outperforms compared other competing methods (Figure 4g-i and the corresponding text). Still, we would like to point out that having TF binding priors largely enhance the performance, which underscores the importance of using prior information that many methods do not make the most of.

3. Single-cell data is ever-increasing, and thus the application of the proposed method for GRN inference could be significantly broader if it could be generalized into single-cell data. Although the authors did not mention this specifically in the manuscript, the gene-by-sample expression matrix here could be easily generalized into a gene-by-cell expression matrix for single-cell data. If combining with other single-cell data analysis methods such as Seurat or scdiff, the authors could generate a gene-by-cell expression matrix for each cell population (i.e., cell type or subtype). Then the proposed model could be extended for cell-type-specific GRN reconstruction, which will immensely broaden the application of the proposed NetREX-CF method. I suggest authors add a single-cell section

: We appreciate the reviewer’s point. The suggestion was also made by other reviewers, from which we could greatly improve our study. In the revised version of our manuscript, we have demonstrated how well NetREX-CF performs in single-cell RNA-Seq analyses. Please see our

response to Reviewer 1 (Main point 1-A), and the new section about the single-cell RNA-Seq in the manuscript (page 17 section 2.4). Thank you.

4. Also, all the validations/valuations were done in the fruitfly, which might be a bit limited as the GRN is much simpler in fruitfly compared to other more complex organisms such as humans or mice. Therefore, to demonstrate the method's performance in complex organisms, I would suggest authors also perform benchmark analysis following a similar approach as discussed in this 2020 nature methods paper (<https://www.nature.com/articles/s41592-019-0690-6>).

: We understand the reviewer's concern, and thus included results from human studies, and performed benchmark analyses according to the reviewer's suggestion (page 17 section 2.4).

In addition, it must be noted that our use of the Drosophila system is not only because it has a streamlined gene network, but also because it provides an excellent high-throughput capability that we needed for the generation of the "new gold standard network" that can replace, or supplement, the yeast gold standard network that the community has been exhaustively using. We hope that our dataset could be widely used by other GRN inference studies. We emphasized this point better in our revised manuscript (page 20).

Minor comments:

1)The Github page provided is NOT publically available. It requires login information. I can't test run the proposed method.

: We apologize for the inconvenience. We have fixed this issue.

2) On page 10, the first paragraph, the authors mentioned that NetREX-CF achieves the lowest overall average ranking score for all but one. I did not find any other methods that perform better in all 4 tasks.

: We are sorry for the confusion. We revised the sentence into "To sum up, NetREX-CF achieves the lowest, thus the best, overall ARS".

3) On page 13, "although the performance of NetREX-CFis", it is missing a space between NetREX-CF and is.

: We apologize for the typo. We have corrected this and others in the manuscript. Thank you.

4) For the equation on page 6, it shouldn't be $-H(S,A)$ and $-F(S,X,Y)$ as the authors are searching for a solution that minimizes the penalized reconstruction error $\|E-SA\|^2$. I guess "-" is not a sign there. I would suggest either removing it or changing it to another symbol to avoid confusion.

: We apologize for the confusion. In the previous manuscript, "-" was not a minus sign but a dash bullet symbol. To avoid confusion, we changed it to a bullet symbol (\bullet), in our revised version.

(5) On page 1, "this approach becomes the foundation of the current state-of-the-art methods for GRN reconstruction". There are a lot of widely used GRN methods (e.g., GENIE3) that are not based on NCA. A lot of methods are based on regression/correlation. Many others are based on probabilistic graphical models. In each of these categories, we all have good methods that are used widely by the GRN community. It is hard to say which methods are the state-of-the-art ones. I would suggest authors avoid claiming the NCA is the foundation of the current state-of-the-art methods.

: The reviewer's suggestion is valid. We have changed the sentence to "this approach becomes the foundation of the current state-of-the-art **NCA-based** methods for GRN reconstruction" (page 1).

(6) In the supplement file, the authors are missing a lot of references (?). There might be other minor issues regarding the supplement. Please take another look and correct all these text/references issues.

: Thank you so much for the catch! We have added the references and thoroughly proofread the manuscript.

Reviewer #2 (Remarks to the Author: Reproducibility):

N/A

: Thank you for reviewing our manuscript!

Reviewer #3 (Remarks to the Author: Overall significance):

This work reports a new computational method for gene regulatory network (GRN) inference from expression data along with prior knowledge of TF-gene regulatory relationships. It presents a methodological advance over the state-of-the-art, specifically by modifying the authors' previously published NetREX method (Nature Comm 2018) to better utilize the prior knowledge. A major contribution of the study is its generation of a large expression data set in *Drosophila* S2 cell lines. Such a data set may be of use to the community in the future.

Despite its positive aspects, the work falls short of convincingly demonstrating the advantage of the new method. Comparisons are made to several existing methods, using one "gold standard" GRN from yeast, but there are concerns about this evaluation. Moreover, the evaluation is on one data set only, out of necessity, since gold standard GRNs are not readily available, but this limitation makes interpretation of perceived improvements difficult. Comparisons are also made using a new "ground truth" constructed using the new data in S2 cells, but these also have significant issues, primarily the fact that the RNAi-based network is dominated by indirect regulatory relationships that are not bona fide GRN edges.

: We thank the reviewer for the summary and helpful comments. We understand the reviewer's concern about the gold standard GRNs. As the reviewer has pointed out, our study initially focused on the yeast gold standard GRN, which is the only available and generally accepted, ground truth in this field. In order to overcome this limitation, we have generated the RNAi-based gene network perturbation set, which is specifically designed for this study. This fact, by itself, should be a significant improvement compared to the previous studies that relied on only the yeast dataset.

We are confident that this will be one of our community's best datasets for the study of GRN inference. Indirect effects are unavoidable for any kind of loss-of-function study, including RNAi, CRISPR, and even drug treatments. Therefore, we presume that the reviewer is most interested in how to minimize the effect. First, we have incubated our cells with RNAi reagent for only "one day". Although the short incubation can minimize any indirect effect, one might worry that the short incubation would lead to incomplete protein-level degradation when the half-life is long. The knockdown efficiency at the RNA level was excellent (Supplementary Figure 1d); thus, the target TF is expected to be reduced by more than 50% at the protein level, considering the doubling time of *Drosophila* S2 cells (~ 22 to 24 hours). Second, in addition to the indirect regulatory effect, we also carefully evaluated potential off-target effects that the RNAi reagents might cause. We observed only a negligible effect as summarized in Supplemental Figure 1e.

In light of this, the work's demonstrable strengths are limited to a technical innovation in the GRN inference methodology and the reporting of a new bulk RNA-seq data set. The paper is well written and to-the-point.

: Thank you.

Major comments:

1. The first main results are shown in Figure 2. Panels A and B compare the accuracy of recovering gold standard GRN edges among different methods. Overall, NetREX-CF (the presented method) outperforms all competing methods in the per-gene comparison and is about equal to one competing method in the per-TF evaluation. A few comments about this evaluation though: **(a)** GENIE3 is missing as a competitor. It is a popular method for expression-based GRN inference, even though it does not use prior information. It is a worthwhile comparison to add, to convince readers. **(b)** The way the results are presented is not entirely satisfactory. It is not clear for example if the "top ranked" genes are systematically biased towards those with fewer regulators (in the gold standard). More commonly, such evaluations are done on a global ranking of all GRN edges, along with ROC and PR curves. Even for per-gene comparison, it may be more meaningful to present per-gene AUROC and AUPRC values as distributions. **(c)** In the current form, it is not clear if it is useful to have a better predictor at a point on the curve where the average rank of "true TFs" is around 10% or worse. A biologist is

unlikely to be interested once the accuracy has dropped to this level, so a performance gap in that range is not particularly helpful.

Response to (a): We thank the reviewer for this comment. In the revised manuscript, we have included GENIE3 as well as GRNBoost2 in all our analyses (Figures 2, 4, and 5). We demonstrated that NetREX-CF outperforms both methods in addition to what we had already demonstrated in our first submission. The advantage of using prior information has been reported in our previous study [2] where we compared the original version of NetREX and GENIE3. We revised our manuscript to highlight these consistent observations (page 17).

Response to (b): We understand the reviewer's concern since we agree that the average rank score would have smaller values when a gene has fewer regulators. However, we would like to underscore that this does not create any problem during the comparisons with other methods for a given TF. It is because, for a specific gene with a fixed number of "gold standard" regulators, the only way to achieve a low average rank score is where the ranking of the "gold standard" regulators is low, which is fair for all the GRN inference methods that we tested.

The reason why we prefer the Average Rank Score (ARS) over AUROC/AUPR is that we want to evaluate whether the rankings of the "gold standard" are recovered. ARS uses the ranking of the edges from the gold standard, which is extremely beneficial when the incompleteness of the gold standard is considered. We know the "gold standard" only contains a part of the true-positives. When computing AUROC/AUPR, all the edges that do not exist in the "gold standard" are treated as true-negatives; this is not appropriate. In contrast, when computing ARS, we do not take the edges that are not in the gold standard into account. It should be noted that ARS is a widely used metric; the original paper was cited approximately 3,000 times [18]. We addressed these points in the discussion section (page 21).

Response to (c): While we understand the reviewer's concern, we do not agree that the edges whose ranking is beyond a certain ad hoc cut-off would have less meaning to biologists. It might be the case for traditional molecular biologists who are interested in the most confident interactions, which would show a strong signal with the blotting techniques. However, people can undergo single-cell RNA-Seq analyses these days, in which a subpopulation-level regulation is important. We believe that low-ranked edges will provide useful information for such analyses, and thus, is of interest to genomicists.

Furthermore, we believe that our prediction is still biologically meaningful even when the ranking of the "true TFs" is larger than 10%. This is because the "true TF" set in the "gold standard" is highly likely incomplete. Therefore, our predictions that are not in the "gold standard" are also worth checking by the biologists.

2. Figure 2b suggests that the NetREX-CF method is mostly driven by the prior networks, considering its performance is so closely matched to PriorSum. Is this true?

: Yes. In Figure 2b, the prior includes TF binding data from ChIP-Seq analyses, which is solid evidence for direct regulation. Therefore, it is expected that the inferred GRN is similar to the PriorSum. In addition, we would like to point out that, different from PriorSum, NetREX-CF can make predictions for the rank of the TF-gene regulations that are not in the prior, which is the major achievement of NetREX-CF (Figure 4 g-i).

3. Figures 2c,d show that the performance advantage of NetREX-CF may be present even on GRN edges not included in prior information. This is promising. The three comments (a,b,c) made above apply here also. Furthermore, it is unclear if the edges present in the gold standard but not in the prior networks are an unbiased and representative “test set”. In my admittedly crude understanding of how the “gold standard” network was constructed originally (and included in the YEASTRACT database), the information represented by the prior networks played an important role, so if an edge is present in the gold standard network but not in the prior networks, it is possible that the edge is actually less reliable. The authors should comment on the possible ascertainment bias introduced in the particular evaluation strategy adopted in Figures 2c,d.

: We are grateful for this helpful comment! The gold standard TF-gene regulations we extracted from YEASTRACT have both DNA binding and expression evidence. Therefore, we believe they are reliable. However, what the reviewer said is reasonable, and may represent the current limitations in using the yeast dataset as the sole source of evaluation basis. This is why we used another set of the gold standard based on Drosophila S2 cells, whose edges can be further cross-validated by genetic tests using animals or by utilizing high-throughput genomics datasets from modENCODE and others. We expanded our discussion section to describe the limitation of the yeast gold standard as the reviewer suggested (page 21). Thank you for the suggestion.

4. Figure 2 is the only systematic evaluation provided in support of the new method. This is because GRN inference evaluation, especially one where prior knowledge is utilized, requires a fairly complete real GRN, and such gold standard networks are simply not available. So the authors understandably rely on the one gold standard network that has been used for evaluation for years now. However, this reviewer believes that such minimal evaluations (on one data set only) do not make a convincing enough case for a new GRN inference method, and perhaps more work is needed in the community to develop methods for GRN evaluation before methods can be comprehensively assessed and compared. Lacking that, works such as the current work can be considered to have a more limited, technical value.

: We fully agreed with the reviewer’s point of view. Indeed, the yeast datasets have been extensively used as the testbed, as the reviewer pointed out; however, evaluating any GRN inference method using the only case would not produce persuasive results. Therefore, we have generated the Drosophila RNAi set, which can replace or supplement the yeast gold standard. We highlighted this importance in the revised version of our manuscript (page 21). As mentioned in our response to point 3, we believe that Drosophila S2 cells are suitable for the future evaluation of GRN inference studies because of that system’s abundant resources; any predictions from gene network modeling can be further tested by cellular- and animal-level

studies. We hope that our dataset may be widely used by the community for future GRN studies. We thank the reviewer for raising this issue.

Furthermore, in the updated manuscript's section 2.4 on page 17, we assess our technique using scRNA-Seq data, giving a total of three examples. With this contribution, we expect to strengthen the case for our study.

5. **(A)** A substantial contribution of the study is the large set of RNA-seq data measuring the transcriptomic effects of ~500 gene knockdowns, representing “all expressed” TFs in S2 cells in *Drosophila*. This data set can be of great value for future studies. However, its use as a validation/test set for GRN inference is questionable, despite its comprehensive nature. This is because one expects the majority of edges in the generated “RNAi network” to be indirect regulatory relationships. An examination of the yeast network should indicate to the authors roughly what fraction of genes “downstream” of a TF (i.e., potentially responsive to the TF’s knockdown) are indirect targets, and I believe that the authors will find this fraction to be large. Thus, it is hard to read too much into the evaluations of Figure 4. Comments made above in the evaluation of the yeast network apply here too. **(B)** For instance, if one is interested in counting the target genes for which the “true TFs” are within the top 5-10% (this itself is rather liberal, since it reports ~25-50 TFs, of which presumably a handful are true), NetREX and NetREX-CF have similar performance (Figure 4a, restricted to the left-most part of the curves) and the advantage of the “CF” part, the main innovation in this work, is less clear.

Response to 5-A: Thank you for the comment. Technically, it is impossible to completely distinguish indirect effects from any of the loss-of-function studies. This is why the ChIP prior is important, and all the methods would better utilize the information. We have explained this in our response to Reviewer 3’s overall significance part (above). We also discussed this concern in greater detail in our revised manuscript (page 17).

Response to 5-B: The advantage of the “CF” part of our new method is clear when NetREX-CF is compared to the original NetREX without CF (Figure 4a). We revised our manuscript to highlight that NetREX-CF outperforms its previous version by a large margin, and underscored the contribution of CF. Thank you.

Minor comments:

Grammatical error: “While, Inferelator [3], a method built on network component analysis (NCA), uses given gene expression data and a network prior to estimate TF activity.” Please check this sentence.

Typographical error: page 3: “To validate the GRNs bluit by ..” Replace bluit with built.

Typo: page 6: “matrix C_{ij} is built form the prior information” ... Replace form with from.

Typo: page 14: “defined by each algorithms. we run Gene Set Enrichment Analysis” Replace period with comma.

Typo: page 14: “in this study together whi collected data” ... Replace “whi” with “with”.

Typo: Acknowledgment to “Alireza Fotuhi Siahpiran” misspells the last name.

We apologize for the typos and grammatical mistakes. We have corrected the above errors, and have also carefully proofread the manuscript during our revision. Thank you so much for helping us improve our manuscript.

Response to Reviewers

: We would like to thank our reviewers for their invaluable comments on our revised manuscript. We found the reviewers' comments to be extremely helpful in further improving the manuscript. In our point-by-point response, we addressed each comment the reviewers made. We used blue text for the reviewers' comments, and black for our responses. In our revised manuscript, we used red text for any changes from the previous version. We hope you will find our manuscript suitable for publication.

Reviewer #1 (Remarks to the Author: Overall significance):

Precisely constructing GRN is an important task. This new method provides another useful tool. The authors compared with the existing methods and showed superior performance on multiple benchmark data sets. The large scale knockdown in S2 cells provides a new benchmark that is useful for the community.

Reviewer #1 (Remarks to the Author: Impact):

The idea of using CF to recommend edges and joint optimization with NCA analysis is interesting.

: We are happy that the reviewer found our study interesting! Thank you.

Reviewer #1 (Remarks to the Author: Strength of the claims):

The authors have addressed most of my previous comments. The revised manuscript is quite improved from the original one.

1. The authors misunderstood a previous comment

“A useful application of GRN is to uncover important regulators in a given cell type. Can NetREX-CF outperform the existing methods on this application? This point is not discussed in the manuscript.”

The authors compared the GRNs constructed by different methods. But is there any way to find which TFs are the most important in a specific cell type? For example, using the GRN in hESC, can NetREX-CF find the TFs crucial for pluripotency given the predicted GRN in hESC? This is not the focus of this work but it is worth of some discussion.

: To the best of our knowledge, most previous works use the fraction of the correctly predicted TF-gene pairs to show the competitiveness of their method. However, such a measure does not assess which method better predicts regulators for a given gene. In this study, we propose a

new rank-based metric that is gene-centric. Using the new metric, we can evaluate whether the predicted regulators for a gene are consistent with the benchmarking data. As can be appreciated from Figure 2a, this is a much more challenging task, and our method outperformed competing approaches by the new metrics. This gives us confidence that our method is very competitive in predicting regulators in general. From our experiments for scRNA-Seq, we demonstrated that we could pinpoint predicted cell type-specific regulators based on cell type-specific gene expression. However, we have no objective test to judge whether NETREX_CF is more successful than others in identifying “important” regulators.

2. Application of NetREX-CF to scRNA-seq data is exciting. More details should be provided such as whether NetREX-CF deals with heterogeneity of the cells or cell subtypes, how to handle the sparsity issue of scRNA-seq, any advantage of using scRNA-seq compared to bulk or pooling together single cells. This new application can significantly enhance the usefulness of NetREX-CF but it also requires additional effort of careful benchmark using different scRNA-seq data sets like what Pratapa et al., Nature Methods, 2020 did. Probably the authors can consider leaving it out for another study.

: To address the sparsity issue of scRNA-seq, we eliminated cells with less than 500 genes expressed and genes that are expressed in fewer than 10% of the cells (Supplemental Materials Section F.). The hHEP and hESC datasets are from a time-course study, which consists of relatively homogenous populations. However, we found that the reviewer’s recommendation for a new benchmark study that will concentrate on cell heterogeneity and data sparsity is extremely helpful. We appreciate the suggestion and will cover the topic in our future study.

Reviewer #1 (Remarks to the Author: Reproducibility):

The source code is accessible from GitHub.

: Thank you for your review! We provided a new GitHub address for better accessibility of the source code (https://github.com/EJIUB/NetREX_CF).

Reviewer #2 (Remarks to the Author: Overall significance):

Overall, the authors pretty much addressed most of my major concerns. Still, there are some relatively minor comments.

Specifically,

(1) For my first major comment on experimental validation, the authors did update the text to explain the potential reasons why many benchmarked methods have inferior performance

compared to random models. The author also revisited the so-called gold standard results that were used to benchmark all the results.

: We are glad to hear that our revised manuscript has addressed the reviewer's concern.

(2) The authors added another set of results (without using the TF prior) to address my second major comment. As shown in figure 4, yes, I am convinced that the method NetREX-CF indeed has the best prediction accuracy (the smallest average rank score). However, I noticed that the GENIE3 method presents a very comparable performance (~ 2% difference) to the NetREX-CF method (without using the TF binding prior). Under such a scenario, I am curious about the running time and memory cost for the NetREX-CF method. If the running time for NetREX-CF is significantly higher compared to GENIE3 (which I know is very efficient), then NetREX-CF is more suitable for the cases when the TF binding prior is available. For datasets without TF binding prior, the model performance will be similar to GENIE3, and users could use the latter to reduce the cost of running resources. In this case, another simulation on partial TF binding prior (e.g., randomly drop TF binding priors) will become necessary to demonstrate the superiority of the method.

: NetREX-CF is computationally more efficient than GENIE3. NetREX-CF directly works on the adjacency matrix of the GRN and simultaneously identifies TFs for all genes. In contrast, GENIE3 predicts TFs for one gene at a time. Use the hESC data as an example, on a computer with a 2.4 GHz CPU and 16 GB of memory, NetREX-CF takes around 20 mins to run. However, the original GENIE3 python code (<https://github.com/vahuynh/GENIE3>), which uses one thread, takes about 12 hours. GENIE3 can be accelerated by parallel computing (<https://github.com/cs205-genie3-parallel/genie3-parallel>). However, when using four threads, GENIE3 (parallel version) takes around 3 hours to run on hESC data. In conclusion, we believe if we have prior knowledge (with/without TF binding prior), NetREX-CF should be the preferred solution.

(3) I am happy to see that the authors did add a single-cell application of the method. I believe that this could significantly boost the application of the toolset to a much broader domain, particularly the rapid-developing single-cell field. This should also help improve the interest of this work to the readership. The authors added another human dataset, which addressed my fourth concern.

: We are pleased to learn that the reviewer's concerns have been addressed in our updated paper.

(4) Again, this is embarrassing. The authors mentioned that they addressed the GitHub accessibility issue. However, it is still not working (https://github.iu.edu/yijwang/NetREX_CF) on my side. It requires login information. Without the Github page, I can't really test and verify the results of the method.

: We are really sorry (and embarrassed) that the GitHub repository was not accessible. We provide a new GitHub address: https://github.com/EJIUB/NetREX_CF. We tested and confirmed that the link is open to the public.

Reviewer #3 (Remarks to the Author: Overall significance):

All but one of my concerns are appropriately addressed in this revision. I have one minor concern remaining, which I leave to the authors' discretion.

My concern regarding the new RNAi-based GRN as a “ground truth” had been that such a network would have an unknown (and large) fraction of indirect edges, i.e., TF1 -> TF2 -> G regulatory relationships being incorrectly noted as TF1->G relationships in the ground truth. The authors respond with “Indirect effects are unavoidable for any kind of loss-of-function study, ...”, which I agree with, and this is the reason, in my opinion, that ground truth GRNs should not be constructed primarily from lof studies; the Maclsaac yeast GRN for instance is based on ChIP seq + motif matches + motif conservation, reflecting direct TF-gene relationships. The authors also state that “the short incubation can minimize any indirect effect ...”, this does not provide any clues about whether such “minimization” has pushed the indirect relationships (which are bound to be much larger due to the large fan-out of GRN nodes) to being small enough to ignore. Incubation for “only one day” is unclear in this regard – do the authors mean that this experimental step does not provide enough time for targets of the knocked-down TF to exert their respective regulatory effects? My understanding was that timescales of feed-forward regulatory action can be as small as minutes and hours. To what extent does the “one day” incubation period prevent such indirect effects of the TF knock-down?

: We thank the reviewer's insightful comments on this matter. However, as the reviewer pointed out, the dynamics of the knockdown effect are difficult to measure accurately especially when a large number of TFs were considered, as in our study. Thus we have changed our sentence into -

“Therefore, we have incubated our cells with RNAi reagent for only “one day” *to minimize the prolonged indirect effect.*”

I do value the new *Drosophila* S2 cells RNAi data set, it can be very valuable for researchers in a variety of ways. I just don't believe that the RNAi-based GRN should be presented as a “gold standard” for evaluation of GRN methods. This network does not intersect expression information with evidences of direct regulation, and while the authors correctly point out that this supports the use of “priors” in GRN inference methods, it also undermines the designation of that network as a gold-standard to evaluate methods on.

: This is a valid concern, and we thank the reviewer for this critical comment. We, therefore, changed the word “gold standard” into “benchmark” or “benchmarking dataset” throughout the manuscript. Thank you.

Response to Reviewers

Reviewer #1 (Remarks to the Author: Overall significance):

While identification of important regulators and application to single cell data are not the focus on the manuscript, the authors may want to have a concise version of their response in Discussion to discuss these issues.

: We have summarized our responses and put them in the Discussion section in the manuscript.

Reviewer #2 (Remarks to the Author: Overall significance):

The authors did address most of my comments in the previous round of revision. Specifically, the authors benchmarked the running time with other GRN methods (i.e., GENIE3) and demonstrated superior performance. Nevertheless, the GitHub page that the authors provided looks unfinished to me and still contains errors, which is quite unusual for a manuscript under the 3rd round of revision.

: We have added more descriptions of how to use our python source codes on the GitHub page. Also, we have ensured that the provided source codes are 100% runnable without any errors.

Reviewer #2 (Remarks to the Author: Reproducibility):

Even with multiple reminders in the previous reviews (and from multiple reviewers), the GitHub page remains problematic. First, the author did not provide the users with a detailed description and usage of the tool the authors developed. Second, there are still bugs in the jupyter notebook they provided on the github page, line 14 and line 15, which prevents me from testing the tool further.

: We have added detailed descriptions of how to use our NetREX-CF on the GitHub page. In addition, we have corrected the errors in the Jupyter notebook. All our codes are 100% runnable without any errors.